# Epithelial coxsackievirus adenovirus receptor promotes house dust mite-induced lung inflammation

Airway inflammation and remodelling are important pathophysiologic features in asthma and other respiratory conditions. An intact epithelial cell layer is crucial to maintain lung homoeostasis, and this depends on intercellular adhesion, whilst damaged respiratory epithelium is the primary instigator of airway inflammation. The Coxsackievirus Adenovirus Receptor (CAR) is highly expressed in the epithelium where it modulates cell-cell adhesion stability and facilitates immune cell transepithelial migration. However, the contribution of CAR to lung inflammation remains unclear. Here we investigate the mechanistic contribution of CAR in mediating responses to the common aeroallergen, House Dust Mite (HDM). We demonstrate that administration of HDM in mice lacking CAR in the respiratory epithelium leads to loss of peri-bronchial inflammatory cell infiltration, fewer goblet-cells and decreased pro-inflammatory cytokine release. In vitro analysis in human lung epithelial cells confirms that loss of CAR leads to reduced HDM-dependent inflammatory cytokine release and neutrophil migration. Epithelial CAR depletion also promoted smooth muscle cell proliferation mediated by GSK3β and TGF-β, basal matrix production and airway hyperresponsiveness. Our data demonstrate that CAR coordinates lung inflammation through a dual function in leucocyte recruitment and tissue remodelling and may represent an important target for future therapeutic development in inflammatory lung diseases.

In the airway, a simple layer of polarised epithelial cells provides a physical protective barrier as well as the primary defence response to environmental insults through constant communication with the immune system[1,2]. There is increasing recognition that interaction of components of the immune system and epithelial cells contribute to driving pathological process in a wide range of respiratory conditions[3,4]. Epithelial cells line the respiratory tract with the primary function of protecting the airway tract from potential pathogens, infections/injury, and to facilitate gas exchange. The epithelium in trachea and bronchi is pseudostratified and primarily consists of three main cell types: cilia, goblet and basal cells. Ciliated cells facilitate the movement of mucus across the airway tract and

goblet cells produce and secrete mucous to trap pathogens and debris within the airway tract. Basal cells are progenitors that can differentiate into epithelial cell types in response to injury in order to restore the epithelial cell layer[5]. An intact epithelium is crucial to maintaining lung homoeostasis and this depends on intercellular adhesion, which is regulated by cell junctions on the basolateral surface of the cell. Cell junctions comprise of apical tight junctions and more basal adherens junctions, which are controlled through protein-protein interactions between receptors, including cadherin and Junctional Adhesion Molecule (JAM) family molecules, coupled to cytoplasmic binding partners. E-cadherin, the key adherens junction receptor in epithelial cells, forms $Ca^{2+}$-dependent homodimers

✉ e-mail: maddy.parsons@kcl.ac.uk

in trans with opposing E-cadherin on neighbouring cells. Stabilisation of E-cadherin is in part controlled by interaction with cytoplasmic adaptor proteins; β-catenin and p120-catenin. These complexes act in concert with tight junction proteins, including occludins and ZO-1 to maintain passage of solutes, airway epithelial barrier formation and integrity[6]. Loss of epithelial integrity in response to infection, allergen exposure, pollution and chemical exposure can trigger the onset of airway inflammation and in the case of allergen exposure, contributes to sensitisation[7].

Coxsackievirus and adenovirus (Ad) receptor (CAR), is a member of the JAM family and immunoglobulin superfamily and functions as a cell–cell adhesion molecule through homophilic interactions in trans. CAR was originally identified as a primary receptor for Coxsackievirus and Ad cell binding[8,9]. However, our work and that of others over the past decade has demonstrated that CAR contributes to stabilising epithelial cell–cell adhesions as both an adherens and tight junction associated protein[10–15]. Targeted removal of CAR in the germline causes early embryonic lethality due to heart failure[16–18]. Conditional knockout of CAR in adult mice leads to multiple defects relating to cell–cell adhesion including dilated intestinal tract, atrophy of pancreas and complete atrio-ventricular block[19]. As well as forming homodimers in trans, epithelial CAR has also been shown to bind in trans to other members of the JAM family proteins, including JAM-L, -A and -C that are expressed on the surface of leucocytes including neutrophils and γδ T cells. This facilitates leucocyte transepithelial migration and activation[4,20–22]. Pro-inflammatory cytokines contribute to transepithelial migration of leucocytes and have also been shown to mediate JAM family protein localisation and function[23]. Our recent studies have shown that CAR levels in epithelial cells control junctional protein dynamics and that this is in part regulated through phosphorylation of the CAR C-terminus at S290/T293 by PKCδ[13,14]. We further defined the pro-inflammatory cytokine TNF as a cytokine involved in promoting phospho-CAR in lung epithelial cells in vitro and in vivo and demonstrated that this requires CAR homo-dimerised in trans[15]. These findings, combined with other evidence highlighting the emerging importance of the epithelium as a primary regulator of lung inflammation place CAR as potential regulator of the interplay between epithelial and immune cells. This may in turn function in the development and progression of lung diseases such as asthma and chronic obstructive pulmonary disease (COPD).

Here we present evidence that epithelial CAR responds to allergen-induced inflammation in the lung through dual promotion of immune cell infiltration and tissue remodelling. Our data show that CAR-deficient airway epithelium shows significantly reduced pro-inflammatory cytokine release and peri-bronchial inflammation following House Dust Mite (HDM) insult, coupled with reduced HDM-induced tissue remodelling. Human bronchial epithelial cells lacking CAR show reduced cytokine release upon HDM treatment, and this leads to a dual effect of reducing immune cell migration and suppressing fibroblast and smooth muscle cell proliferation. HDM induces phosphorylation of CAR in vitro leading to increased dynamics of CAR at epithelial cell junctions and dissociation of CAR from the mechano-sensitive binding partner caveolin-1. CAR is required for triggering of cytokine release to promote neutrophil and γδ T-cell recruitment to the lung upon HDM challenge, exacerbating tissue remodelling. However, we also demonstrate that depleting CAR in the airway epithelium enhances epithelial TGF-β activation, leading to basal matrix deposition. We postulate that phosphorylation of CAR induced by pro-inflammatory triggers destabilises this receptor at the membrane and induces pro-inflammatory cues from the epithelium, as well as promoting CAR-immune cell interactions to induce tissue-level inflammation. Our data implicate a dual function for CAR in promoting lung inflammation whilst being critical for maintaining lung tissue homoeostasis.

## Results

### CAR promotes neutrophil and γδ T-cell lung infiltration in response to HDM

To explore the function of CAR in lung epithelial homoeostasis, we specifically deleted CAR from airway club cells by crossing Scgb1a1-CreER mice with CAR[fl/fl] mice[19] (Supplementary Fig. 1a). Deletion of CAR in the airway epithelium using Tamoxifen treatment was confirmed by immunostaining of mouse lung sections (Supplementary Fig. 1b) and qPCR of whole lung extracts (Supplementary Fig. 1c). We further confirmed that Tamoxifen treatment alone in C57BL/6 mice did not lead to changes in immune cell profiles in lungs compared to control mice treated with corn oil (Supplementary Fig. 1d, e). To analyse contributions of epithelial CAR to allergen-induced inflammation, we treated control mice (Mock) or lung epithelial CAR knockout mice (Ep-CAR KO) with PBS or HDM for 5 weeks and analysed immune responses in both bronchoalveolar lavage fluid (BAL) and within the lung tissue. HDM induced a robust inflammatory response in the BAL of both Mock and Ep-CAR KO mice (Supplementary Fig. 2a) with no changes in immune cell profiles between the two groups before or after challenge (Supplementary Fig. 2b–d). However, analysis of cell infiltrate into lungs and H&E-stained tissues showed a significant reduction in the overall immune cell infiltration in HDM-challenged Ep-CAR KO mice compared to controls (Fig. 1a, c). We note that whilst Tamoxifen alone did not induce an immune response (Supplementary Fig. 1d, e), no mice in our groups were treated with both Tamoxifen and HDM so we cannot rule out a possibility of a combined effect of these agents in the Ep-CAR KO mouse immune phenotype. Additional staining using Periodic acid-Schiff (PAS) to detect polysaccharides demonstrated a significant reduction in levels in Ep-CAR KO lungs, suggesting reduced mucus production (Fig. 1a, b) indicative of an overall reduction in tissue inflammatory response to HDM.

CAR can heterodimerize with receptors expressed on neutrophils and γδ T cells and this mediates immune cell transmigration. Indeed, profiling of immune cells affected by CAR depletion within the lung showed significant reductions in neutrophils and γδ T cells (Fig. 1d, e). Significantly reduced neutrophil infiltration in Ep-CAR KO animals was also confirmed using Ly6G staining of both frozen lung sections (Supplementary Fig. 2c) and in Precision Cut Lung Slices (PCLS) (Fig. 1f, g). Notably we observed high levels of neutrophils co-locating with epithelial cells lining the airways of Mock animals challenged with HDM (Fig. 1h). To understand whether CAR was paying a direct function in attracting immune cells to the lung, we depleted CAR from 16HBE human bronchial epithelial cells using CRISPR, and subsequently generated rescue cell lines by stably infecting with CAR-GFP (Supplementary Fig. 2d). To understand whether human epithelial cell secretion of pro-inflammatory factors was impacted by removal of CAR, we harvested supernatants from HDM-challenged 16HBE cell lines and analysed the ability of HL60 cells (a characterised human neutrophil model cell line[24]) to chemotax towards the conditioned media in transwell assays. Media from parental 16HBE cells challenged with HDM led to a significant increase in HL60 migration (Fig. 1i). We note that HDM remained within this conditioned media, and whilst unlikely to be highly active we cannot rule out additional contributions from this remaining HDM on neutrophil behaviour in this assay. Notably however, media from CAR CRISPR cells induced a significantly reduced chemotactic response compared to parental cells, an effect that was restored in CRISPR cells expressing CAR-GFP (Fig. 1i). This data combined suggests that CAR expression within the lung epithelium promotes immune cell recruitment through regulation of secreted factors and likely also through direct interactions with receptors on the surface of these cells as previously described.

### Lung epithelial CAR contributes to cytokine secretion

Allergic response is characterised by secretion of Th2 cytokines[25]. To explore whether CAR functions in Th2 responses, we firstly adopted a

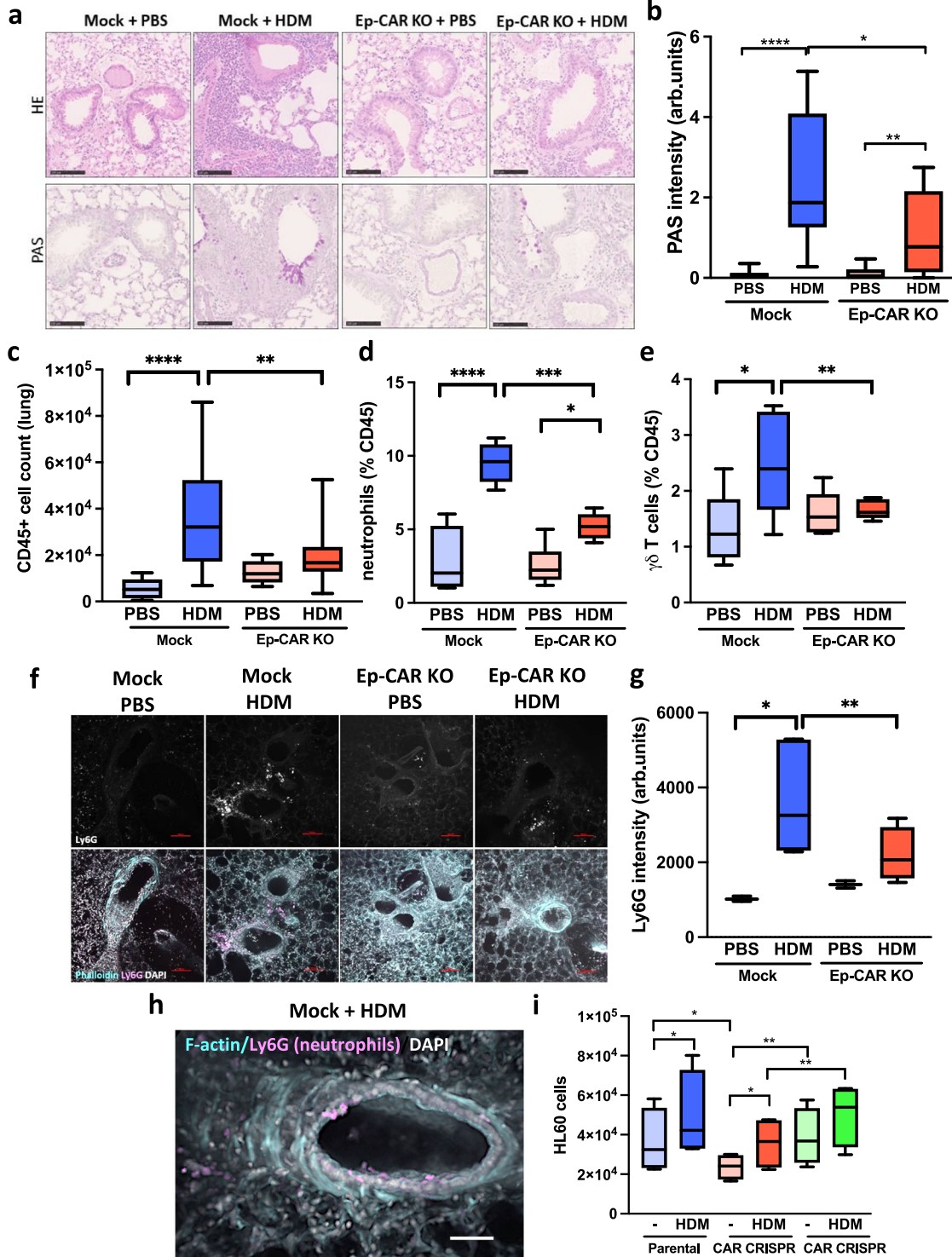

**Fig. 1 | CAR promotes lung inflammation in response to house dust mite (HDM).**
**a** Representative images of H&E (top) and Periodic Acid-Schiff Staining (PAS; bottom) of FFPE lung sections from indicated animals and conditions. Scale bars 100 μm. **b** Quantification of PAS staining from 5 mice per group. For each mouse, one lobe was collected for PAS staining, and 6 airways/mouse analysed; represents 3 independent experiments **c** CD45$^+$ positive cells in lung homogenates from indicated conditions. **d** % of LYC6G + ve neutrophils and **e** % of γδ T cells in CD45$^+$ positive population in lung homogenates. For **c–e**, data is from one lobe per mouse, from 5 mice; representative of 3 independent experiments.
**f** Representative confocal Z-projection of Precision Lung Cut Slices (PCLS). Ly6G staining (upper panels), and merged images shown (actin = cyan; Ly6G =

magenta; DAPI = blue). Scale bar 100 μm. **g** Quantification of Ly6G mean intensity (as in **f** from 5 airways per PCLS from each group (one PCLS per mouse; 5 mice per experiment); 3 independent experiments. **h** Representative confocal Z-projection of an airway from a mock HDM-treated animal (cyan = F-actin; Ly6G = magenta; DAPI = blue)). **i** HL60 cell migration in response to supernatants from specified 16HBE cells ± HDM; $n = 10$ per cell line/condition, 4 independent experiments. All graphs show median (line) with 25/75 percentiles (box) and min/max values. One-way ANOVA analysis with Dunnett's post hoc test was performed to test statistical significance except for (i) where two-way ANOVA with Tukey's post hoc test was used. $P$ values *$p < 0.05$; **$p < 0.01$; ****$p < 0.0001$. Source data are provided as a Source Data file.

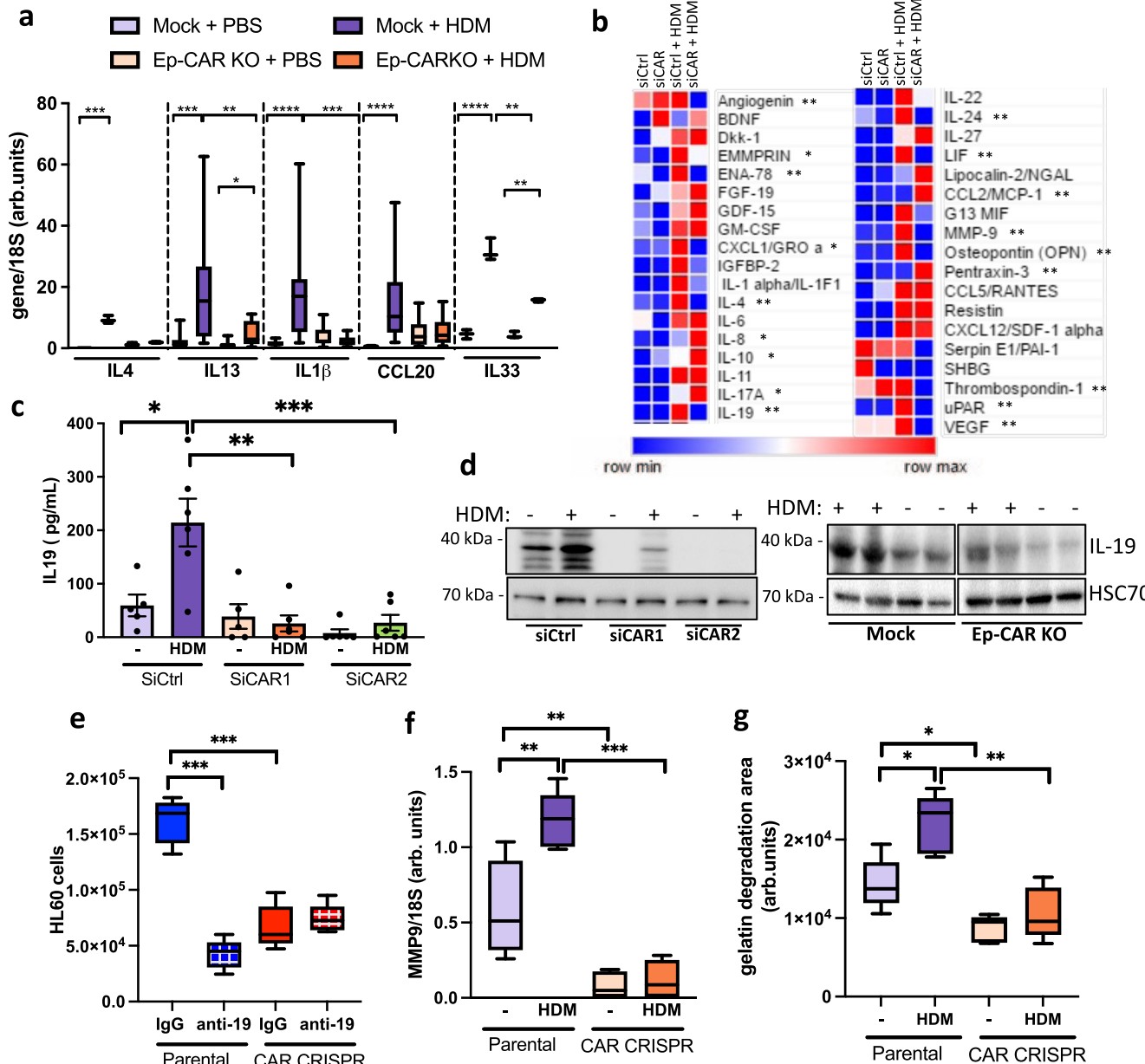

**Fig. 2 | Lung epithelial CAR contributes to cytokine secretion profiles. a** Levels of IL4, IL13, IL1b, CCL20 and IL33 transcripts from lung extracts specified animal groups measured by qPCR. Data is pooled from 3 independent in vivo experiments with 5 mice per group per experiment. **b** Summary of results from cytokine array from 16HBE siControl or siCAR cells ± HDM for 24 h. Asterisks indicate significant changes in CAR depleted cells +HDM compared to control cells + HDM **c** IL-19 levels in supernatants of 16HBE cells ± HDM for 24 h as measured by ELISA. Graph shows pooled date from 6 replicates, representative of 3 independent experiments. **d** Western blots of lysates from 16HBE as in **c** or homogenates from total lungs from Mock or Ep-CAR KO mice probed for levels of IL-19 and GADPH. Representative of 4 experiments for each. **e** Transmigration of HL60 cells in response to supernatants from specified cell lines in presence of control IgG or IL-19 function blocking antibody (anti-19). Data shown from 3 samples per condition; representative of 3 independent experiments. **f** Levels of MMP9 transcripts from specified cells ± HDM for 24 h measured by qPCR. Data shown from 4 samples per condition; representative of 3 independent experiments. **g** Degradation of fluorescently labelled gelatin by specified cells ± HDM. Data is from 15 samples per condition; representative of 3 experiments. All graphs show median (line) with 25/75 percentiles (box) and min/max values. One-way ANOVA analysis with Dunnett's post hoc test was performed to test statistical significance in **a, b**; two-way ANOVA with Tukey's post hoc test was used for **c, e, f, g**. P values *$p < 0.05$; **$p < 0.01$; ***$p < 0.0005$; ****$p < 0.0001$. Source data are provided as a Source Data file.

candidate approach by analysing transcript levels of Th2-related cytokines interleukin (IL)4, IL13, IL33, IL1β and CCL20 in the lungs of Mock and Ep-CAR KO mice challenged with HDM as these have known functions in asthma[25–28]. Mock animals showed a significant increase in all these cytokines following HDM challenge, whereas this response was significantly impaired in Ep-CAR KO mice (Fig. 2a). To assess contributions specifically from the epithelium, we performed cytokine array analysis on supernatants from siControl or siCAR treated cells in culture. Data showed a significant reduction in several secreted factors

in CAR depleted cells challenged with HDM compared to control cells, including CXCL1, IL-1α, IL-10, IL-17, IL-19, IL-24, MMP-9 and LIF (Fig. 2b). Several targets that have been shown to be important in asthma (LIF1, IL-19, IL-4, MMP-10 and −13)[25–28] were subsequently validated by qPCR and ELISA (Supplementary Fig. 3a–c). Of the significantly altered molecules, we considered targets that were differentially regulated in CAR depleted cells upon HDM challenge and focused on IL-19, as recent reports have shown a positive correlation between levels of this cytokine and allergic airway inflammation[29] as well as boosting of

typical Th2 cytokines such as IL-4 and IL-13[30]. Reduced HDM-induced IL-19 levels were validated following removal of CAR from epithelial cells both in vitro and in vivo (Fig. 2c, d). We further demonstrated the functional relevance of reduced IL-19 levels in facilitating the recruitment of neutrophils by treating conditioned media from 16HBE cells with IL-19 functional blocking antibodies. Chemotaxis assays demonstrated a significant reduction in HDM-induced HL60 cell migration in the presence of IL-19 blocking antibodies, whereas the reduced HL60 migration in response to CAR CRISPR media was unaffected (Fig. 2e). Interestingly, data from the array also demonstrated changes in secreted proteins involved in extracellular matrix (ECM) remodelling in CAR depleted cells, including MMP9 or uPAR (Fig. 2b). Reductions in MMP9 transcript in CAR CRISPR lung epithelial cells was confirmed using qPCR (Fig. 2f) whereas levels of other MMPs were unaffected (Supplementary Fig. 3d). The reduced MMP9 activation and secretion in the absence of CAR after HDM treatment verified using gelatin degradation assays, as gelatin is a substrate for MMP9. Levels of degraded gelatin in HDM-treated CAR CRISPR cells were significantly reduced compared to parental cells treated with HDM (Fig. 2g). These findings demonstrate that CAR functions in epithelial cell responses to allergen that in turn lead to immune cell infiltration and tissue remodelling.

## CAR is required for bronchial epithelial monolayer integrity

The airway epithelium forms the first structural barrier against environmental insults including deposited aeroallergens and contributes to initiation of allergic airway inflammation and remodelling. Many aeroallergens, including HDM, are known to lead to altered epithelial cell morphology which can lead to compromised barrier integrity[31,32]. Given that CAR is known to control epithelial cell–cell adhesion, we investigated whether depleting CAR from epithelial cells altered monolayer structure and integrity. Analysis of the epithelial layer in lungs from Mock and Ep-CAR KO mice demonstrated a significant increase in the epithelial cell height in Mock mice treated with HDM compared to PBS treated Mock animals (Supplementary Fig. 4a, b). We also observed a significant increase in epithelial height in Ep-CAR KO lungs under basal (PBS) conditions compared to those from Mock animals, which was not altered by HDM challenge (Supplementary Fig. 4a, b). The average airway diameter was also reduced in Ep-CAR KO mice compared to Mock under basal conditions as measured in ex vivo PCLS following Metacholine treatment (Fig. 3a) indicating higher contractility in the absence of epithelial CAR. To understand whether this altered morphology also corresponded with barrier integrity changes, permeability was assessed through analysis of TRITC-dextran passage across 16HBE cell line monolayers. Depletion of CAR from human epithelial cells resulted in increased basal permeability and reduced barrier integrity, which was restored upon rescue of CAR levels (Fig. 3b). Interestingly, removal of CAR from 16HBE cells also led to reduction in early adhesion to the collagen I and Matrigel (Fig. 3c, d), suggesting crosstalk between CAR and integrin-based adhesions, as we have previously reported in other cell types[11]. Further analysis demonstrated no change in levels of cell–cell adhesion proteins E-Cadherin or ZO-1 upon CAR deletion (Supplementary Fig. 4c), but ZO-1 was mis-localised both in vitro and in vivo (Fig. 3e, f) and this was also the case for E-Cadherin (Supplementary Fig. 4d, e). Combined, this data supports the notion that CAR maintains basal epithelial barrier integrity through maintenance of optimal cell–cell and cell-matrix adhesions.

## HDM treatment leads to phosphorylation and increased movement of CAR at junctions

We have previously shown that the CAR cytoplasmic domain is phosphorylated at T290/S293 which can be triggered by pro-inflammatory cytokines leading to junction destabilisation and leucocyte transmigration[14,15]. In support of our previous findings, analysis of 16HBE parental cells demonstrated increased phospho-CAR (P-CAR) in

the presence of HDM confirmed by Western blotting and immunofluorescence (Fig. 4a, b). This was further confirmed in ex vivo in PCLS from Mock mice treated with HDM (Fig. 4c). We have previously shown PKCδ is the kinase for T290/S293 P-CAR[14], and our analysis also showed that activation of PKCδ was enhanced by HDM treatment (Fig. 4d). To test whether HDM-induced P-CAR alters CAR behaviour in cells, we analysed dynamics of CAR at cell–cell adhesions in live cells treated with HDM. Line scan analysis of CAR perpendicular to the junction demonstrated a significant reduction in CAR levels at cell–cell adhesions following HDM treatment as indicated by more flattened traces of CAR intensity under HDM conditions compared to PBS at the same timepoint (Fig. 4e), consistent with the notion that HDM-induced P-CAR promotes CAR mobility. Importantly, the same analysis of a T290/S293 phospho-dead mutant of CAR (AACAR) demonstrated highly stable and tightly localised distribution of this mutant at epithelial cell junctions, which was unaffected by HDM challenge (Fig. 4e). Moreover, depletion of CAR from 16HBE cells resulted in disruption of cortical F-actin filaments at epithelial cell–cell adhesions in the absence of HDM (Fig. 4f). To test whether CAR homodimerization in trans was required for CAR mobility, 16HBE parental cells were treated with purified Ad Type5 fibre knob (FK) that binds to CAR with higher affinity than CAR binds to itself[33,34]. FK treatment resulted in disruption of CAR at junctions under basal conditions (Fig. 4g) as well as reduced HDM-induced P-CAR (Fig. 4h). These findings demonstrate that CAR is phosphorylated upon HDM challenge, and this leads to disruption of CAR at cell–cell adhesions and concomitant reduced junction stability.

## CAR forms a complex with caveolin-1

CAR has previously been shown to associate with several intracellular proteins that co-ordinate cell–cell adhesion, including MUPP-1, MAGI and LNX[10,35,36]. However, potential interaction partners mediating CAR-dependent functions in the lung epithelium have not been previously assessed in an unbiased manner. To address this, we performed BioID in 16HBE CAR CRISPR cells re-expressing CAR-BirA to enable capture of protein interactions with CAR. Analysis of biotinylated proteins by mass spectrometry demonstrated over 700 CAR-associated proteins, covering many previously reported binding partners (Supplementary Fig. 5a, b; Supplementary Data 1). The most highly represented and enriched protein in the dataset was caveolin-1, a mechano-sensitive scaffolding protein found within membrane microdomains called caveolae and previously reported to stabilise cell–cell adhesions[37–39]. To investigate if caveolin-1 was involved in CAR-dependent HDM-induced epithelial destabilisation, we analysed expression and localisation of caveolin-1 in 16HBE parental, CRISPR and CAR overexpressing cells. There was no change in total protein levels upon CAR manipulation (Fig. 5a), but reduced localisation of caveolin-1 at cell–cell adhesions in HDM-challenged 16HBE cells (Fig. 5b; Supplementary Fig. 5c). Moreover, caveolin-1 was lost from cell–cell adhesions in 16HBE CAR CRISPR cells (Fig. 5b) further suggesting these two proteins co-operate at these subcellular sites. To validate formation of the CAR-caveolin-1 complex, we performed immunoprecipitations of CAR-GFP followed by probing of complexes for caveolin-1. Caveolin-1 was present in a complex with CAR (Fig. 5c) and this was disrupted upon HDM challenge (Fig. 5d). Furthermore, levels of caveolin-1 were higher in complex with AACAR-GFP than WTCAR, and this complex was not disrupted in response to HDM treatment (Fig. 5d). These data demonstrates that caveolin-1 binds CAR and may aide in CAR-dependent responses to HDM that destabilises mechano-integrity of epithelial monolayers.

## Epithelial depletion of CAR results in increased airway contractility

CAR promotes epithelial cell integrity and homoeostasis but also drives destabilisation of barrier function and enhanced immune cell infiltration following HDM-challenge. To understand the function of

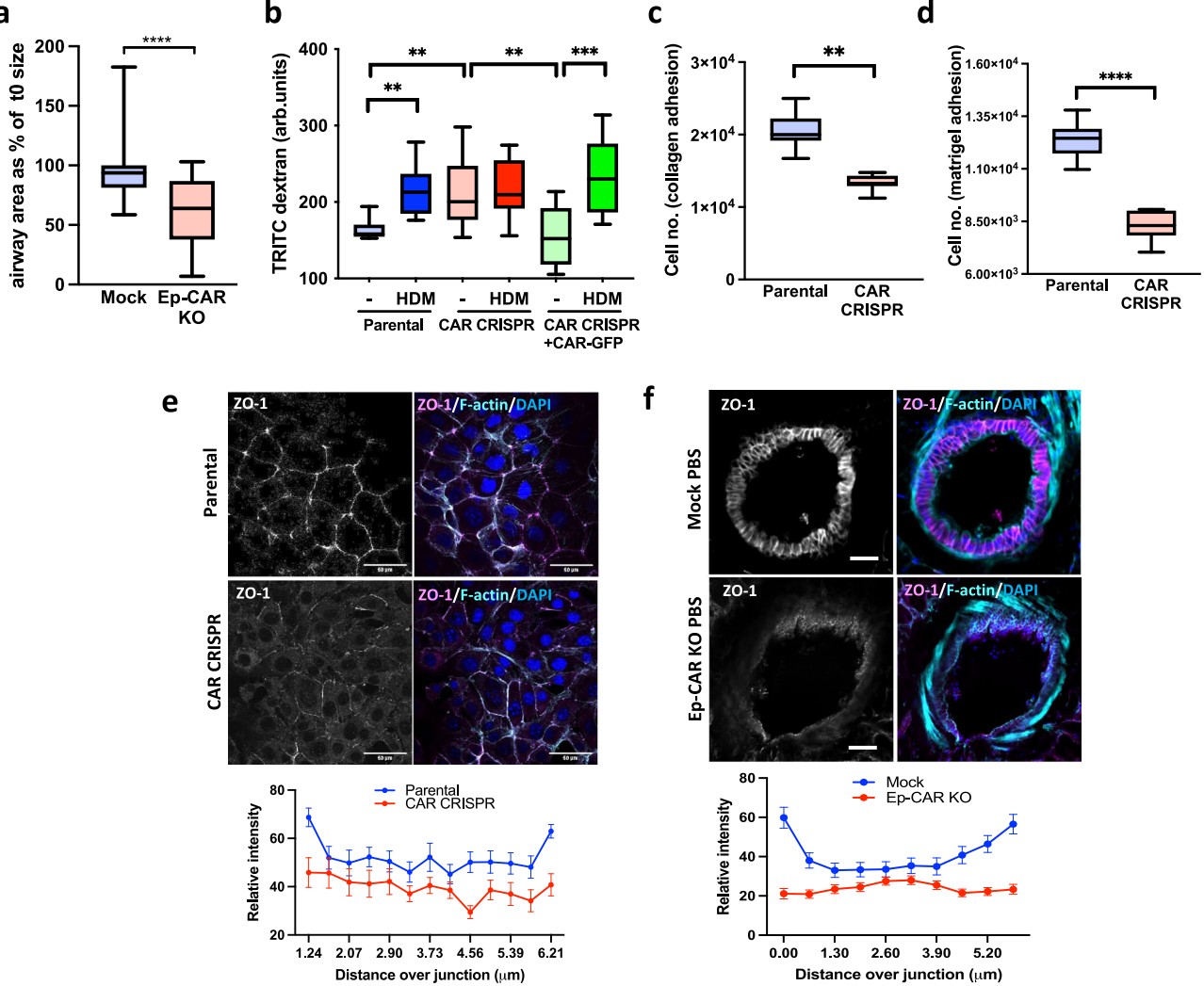

**Fig. 3 | CAR promotes bronchial epithelial monolayer integrity. a** Airway diameter in PCLS from Mock or Ep-CAR KO mice subjected to 0.5 g/ml of Metacholine for 15 min. Graph shows airway area after 15 min as % of starting size from 35 airways per condition from 10 different mice per group (1 PCLS per mouse) representing 2 independent experiments. **b** Permeability assays in specified cell lines showing levels of TRITC-dextran passed through cell monolayers ± HDM. Data is from 3 wells per condition per experiment: representative of 3 experiments. **c, d** Number of adherent cells to collagen I **c** or Matrigel **d** coated wells 2 h post-plating. Data is from 9 wells per condition: representative of 3 experiments. **e** Representative confocal images of specified cells stained for ZO-1 (magenta), F-actin (cyan) and DAPI (blue). Scale bar: 50 μm. Graphs below show quantitative line scan analysis of ZO-1 signal intensity at the cell junctions. Data pooled from 30 cells per condition representative of 3 independent experiments. **f** Representative confocal images of fixed PCLS from Mock or Ep-CAR KO mice stained for ZO-1 (magenta), F-actin (cyan) and DAPI (blue). Scale bar: 50 μm. Data pooled from 50 cells per condition. For each group, 3 PCLS from different mice were analysed; representative of 3 independent experiments. All graphs show median (line) with 25/75 percentiles (box) and min/max values except for those in e,f where means ± SEM are shown. Unpaired 2-tail student's T-tests were performed on data in **a**, **c**, **d**; two-way ANOVA analysis with Tukey's post hoc test was used for **b**. P values **$p < 0.01$; ****$p < 0.0001$. Source data are provided as a Source Data file.

CAR in airway function, we treated Mock and Ep-CAR KO mice with HDM intranasally over 5 weeks and assessed airway contractility using two complementary approaches. Firstly, we treated dissected airways ex vivo with Carbachol to induce contraction and analysed responses using myography (Fig. 6a). Secondly, we analysed PCLS treated ex vivo with methacholine to induce constriction over time (Fig. 6b, c). In both experimental conditions, mice treated with HDM showed increased airway contractility, similar to allergen-induced airway hyper-responsiveness (AHR) seen in asthma[40]. However, airways from Ep-CAR KO mice show increased contractility under basal conditions, which was not further enhanced by HDM (Fig. 6a–c). Previous studies have demonstrated airway smooth muscle (ASM) is important in broncho constrictive airways[41,42]. To understand whether ASM phenotypes may be contributing to the AHR in CAR depleted airways, we quantified α-SMA staining surrounding the airways in lung sections from Mock and Ep-CAR KO mice. Analysis showed a significant

increase in α-SMA signal intensity following HDM treatment in Mock animals, and an increase of the α-SMA levels independently of HDM in Ep-CAR KO lungs (Fig. 6d, e). We further characterised α-SMA organisation in more detail by staining fixed PCLS and quantifying confocal images (Fig. 6f). Results from this analysis demonstrated a significant reduction in distance between α-SMA bundles in regions surrounding the airway (Fig. 6g), and a concomitant increase in α-SMA bundle thickness (Fig. 6h). These data suggest that CAR acts to maintain normal airway epithelial and smooth muscle physiology. To investigate how it impacts ASM, we next investigated whether CAR regulated α-SMA and connective tissue organisation.

### Epithelial depletion of CAR induces TGF-β-dependent airway remodelling

Allergic airway inflammation is associated with increased deposition of ECM proteins within the surrounding connective tissue and airway

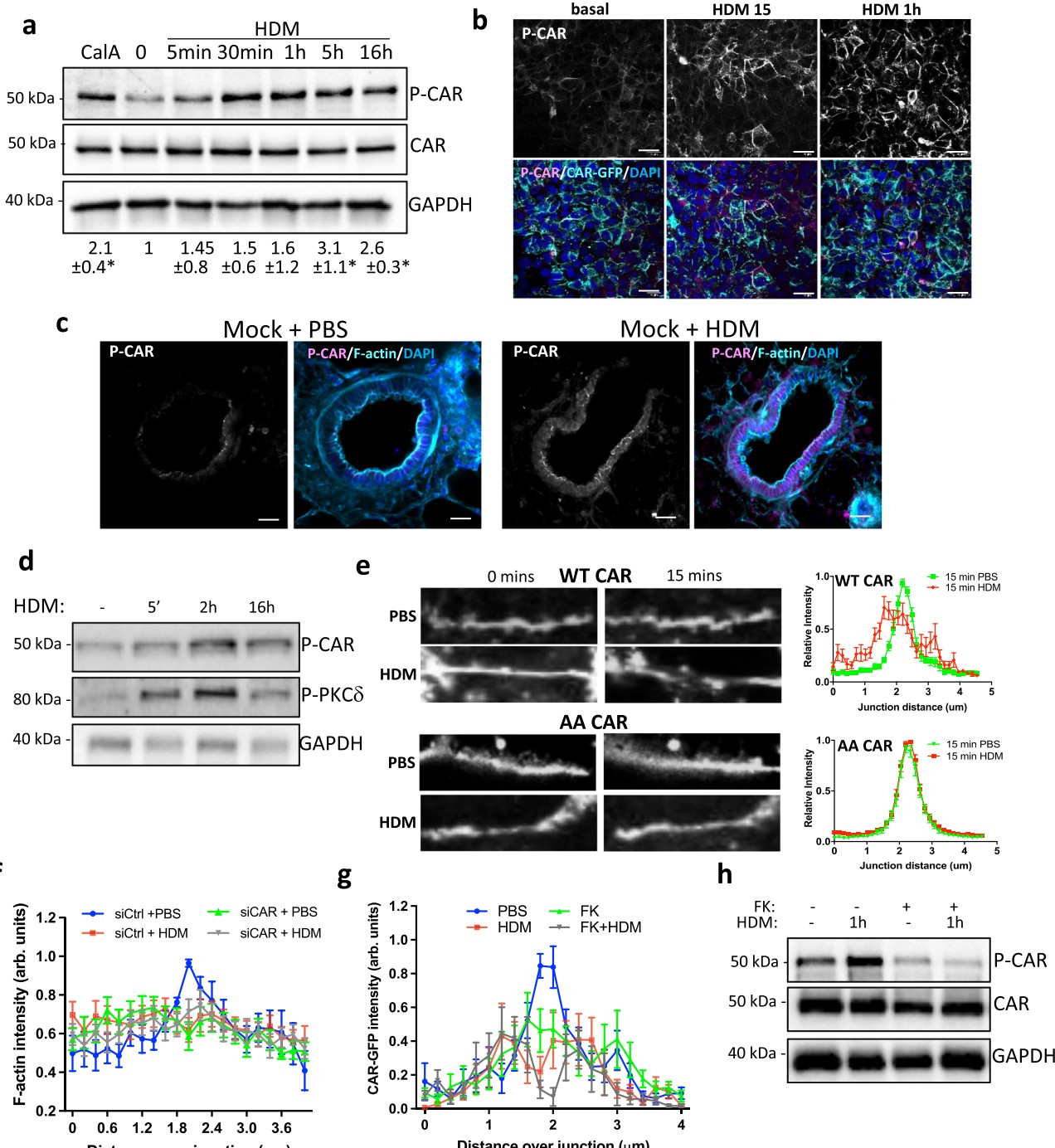

**Fig. 4 | HDM treatment leads to phosphorylation and increased mobility of CAR at junctions. a** 16HBE CAR-GFP cells treated with Calyculin A (CalA) for 5 min or with HDM for the indicated times. Representative western blots probed for CAR, T290/S293 phospho CAR (P-CAR) and GAPDH. Quantification pf pCAR from three independent experiments is shown below. **b** Confocal images of CAR-GFP (green) 16HBE cells treated with HDM as indicated. Cells were stained for P-CAR (magenta) and DAPI (blue). Representative of 3 independent experiments. Scale bars; 50 mm. **c** Confocal images of PLCS from Mock mice ± HDM treatment over 5 weeks, fixed and stained for P-CAR (magenta), F-actin (cyan) and DAPI (blue). Scale bars; 50 mm. **d** 16HBE cells treated with HDM for indicated times, lysed and subjected to western blotting probing for P-CAR and P-PKCδ. Representative of 3 independent experiments. **e** Confocal images of live 16HBE CAR-GFP or AACAR-GFP cells treated with PBS or HDM imaged over 60 min. Images at 0 and 15 min are shown. Line graphs of CAR-GFP and AACAR-GFP at cell–cell junctions for 20 different junctions in each condition, representative of 3 independent experiments. **f** 16HBE cells expressing lifeact-GFP transfected with siControl or siCAR, treated with PBS or HDM and imaged live for 1 h. Graphs show F-actin intensity across junctions for 20 different junctions in each condition, representative of 3 independent experiments. **g** 16HBE CAR-GFP cells treated with Ad5Fiberknob (FK) for 2.5 h followed by PBS or HDM for 1 h. Graphs show CAR-GFP intensity across junctions for 20 different junctions in each condition, representative of 3 independent experiments. **h** Cells as in G lysed and probed for P-CAR and total CAR. All values are mean ± SEM. One-way ANOVA analysis with Dunnett's post hoc test was performed to test statistical significance in **a**. *P* values *$p < 0.05$. Source data are provided as a Source Data file.

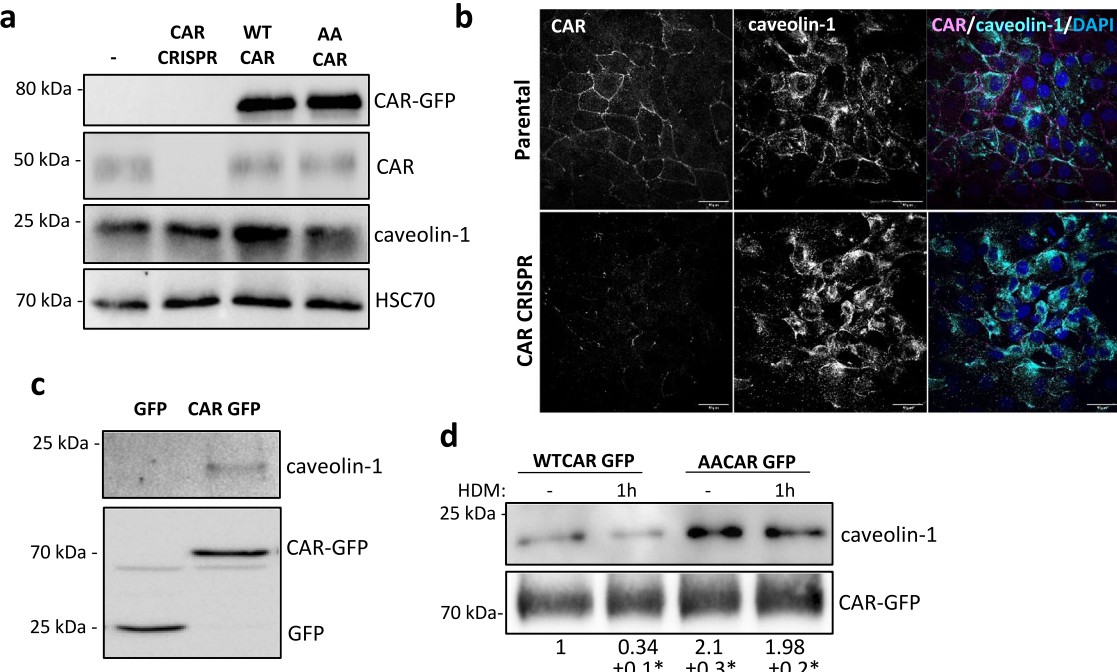

**Fig. 5 | CAR forms a complex with caveolin-1. a** Lysates from 16HBE parental, CRISPR, CAR-GFP and AACAR-GFP cells probed for GFP, CAR, caveolin-1 and HSC70. Representative of 3 independent experiments (**b**) Confocal images of 16HBE parental and CAR CRISPR cells fixed and stained for CAR (magenta), caveolin-1 (cyan) and DAPI (blue). Scale bars: 50 mm. **c** Lysates from 16HBE cells expressing GFP or CAR-GFP were subjected to GFP-trap immunoprecipitation, western blot and membranes probed for caveolin-1, CAR (CAR-GFP) and GFP. Representative of 5 independent experiments. **d** 16HBE cells expressing CAR-GFP ± HDM for 1 h lysed and subjected to GFP-trap immunoprecipitation, western blot and membranes probed for caveolin-1 and GFP. Representative of 5 independent experiments. Source data are provided as a Source Data file.

epithelial cells have been proposed to positively contribute to this pathology[43,44]. To test whether the enhanced contractility in airways in both HDM and Ep-CAR KO mice was due to ECM remodelling, collagen and fibronectin levels were quantified in lungs from mice with and without HDM challenge. Data demonstrated a significant increase in collagen and fibronectin deposition in the lungs of Mock mice treated with HDM (Fig. 7a, b). However, this increase was also evident at basal levels in Ep-CAR KO animals, and levels did not change upon HDM challenge (Fig. 7a, b). To understand how these effects may be mediated, we treated primary human lung fibroblasts and airway smooth muscle cells with supernatants from 16HBE parental and CAR CRISPR and assessed their proliferation over 48 h. A significant increase in proliferation of both cell types was seen in the presence of CAR CRISPR conditioned media compared to parental controls (Fig. 7c). Moreover, Western blotting for collagen I and fibronectin levels in both fibroblasts and smooth muscle cells demonstrated that CAR CRISPR conditioned media stimulated significant increases in deposition of both ECM proteins (Fig. 7d, e). One of the key secreted proteins involved in lung epithelial-stroma interactions and ECM remodelling is TGF-β[45]. We therefore analysed active TGF-β levels secreted by 16HBE parental and CAR CRISPR cells. This analysis demonstrated a significant increase in active TGF-β in CAR CRISPR cells compared with parentals (Fig. 7f), and this was mirrored by increased TGF-β1 transcript levels in these cells, as measured by qPCR (Fig. 7g). We postulated that the increased secretion of TGF-β by CAR CRISPR cells may be responsible for the increased proliferation of fibroblasts and smooth muscle cells as well as increased synthesis of collagen and fibronectin. To investigate this, we repeated these experiments in the presence of a TGF-β receptor type I inhibitor, R268712. Analysis from both experiments demonstrated that blocking TGF-β signalling inhibited proliferation and collagen I production in primary human airway smooth muscle cells (Fig. 7h, i). Collectively, this data demonstrates that depletion of CAR induces increased TGF-β levels that in turn drive matrix remodelling and potentially promote basal airway constriction observed in Ep-CAR KO mice.

## CAR controls a GSK3β-TGF-β signalling axis in lung epithelium

Our data demonstrates that TGF-β activation is enhanced in the absence of CAR, but the pathways upstream of this remain unclear. To define potential signalling changes in CAR depleted cells, we adopted an unbiased approach by assaying activity of >40 human kinases in lysates from 16HBE parental and CAR CRISPR cells with and without HDM treatment. Several targets were differentially regulated depending both on the expression of CAR and the presence or absence of HDM (Supplementary Fig. 6a, Fig. 8a). Some key targets of interest to the phenotypes observed in CAR depleted cells were subsequently validated by western blotting including pSTAT1, ERK1/2 (Supplementary Fig. 6b, c) and GSK3β (Fig. 8b, c). Of particular interest was GSK3β, which has been shown to be involved in both airway smooth muscle hypertrophy and fibrosis in the airway[46,47]. Phosphorylation was increased at Ser9, which inactivates GSK3β, in CAR depleted cells, both in vitro (Fig. 8b, c) and ex vivo in PLCS (Fig. 8d)[48]. GSK3β activation has been suggested to act downstream of TGF-β in promoting fibrosis[49]. However, recent evidence suggests that GSK3β can undergo inactivation by extended exposure to TGF-β[50] leading to stabilisation of SMAD3 and increased TGF-β release through a positive feedback loop and subsequent fibrosis[51]. This suggests convergence of these pathways are more complex in fibrosis than originally suggested[52]. Our analysis of both 16HBE cell in vitro and PCLS ex vivo demonstrated a significant increase in pSMAD2/3 following CAR depletion (Fig. 8e–h). To investigate the involvement of GSK3β in this, cells were treated with the GSK3β inhibitor CHIR-9902, which led to enhanced TGF-β production in parental 16HBE cells, mimicking the phenotype seen in CAR CRISPR cells (Fig. 8i). GSK3β inhibition of 16HBE parental cells also led to an accumulation of β-catenin as previously reported[53,54], concomitant with an increase in pSMAD2/3 (Supplementary Fig. 6d).

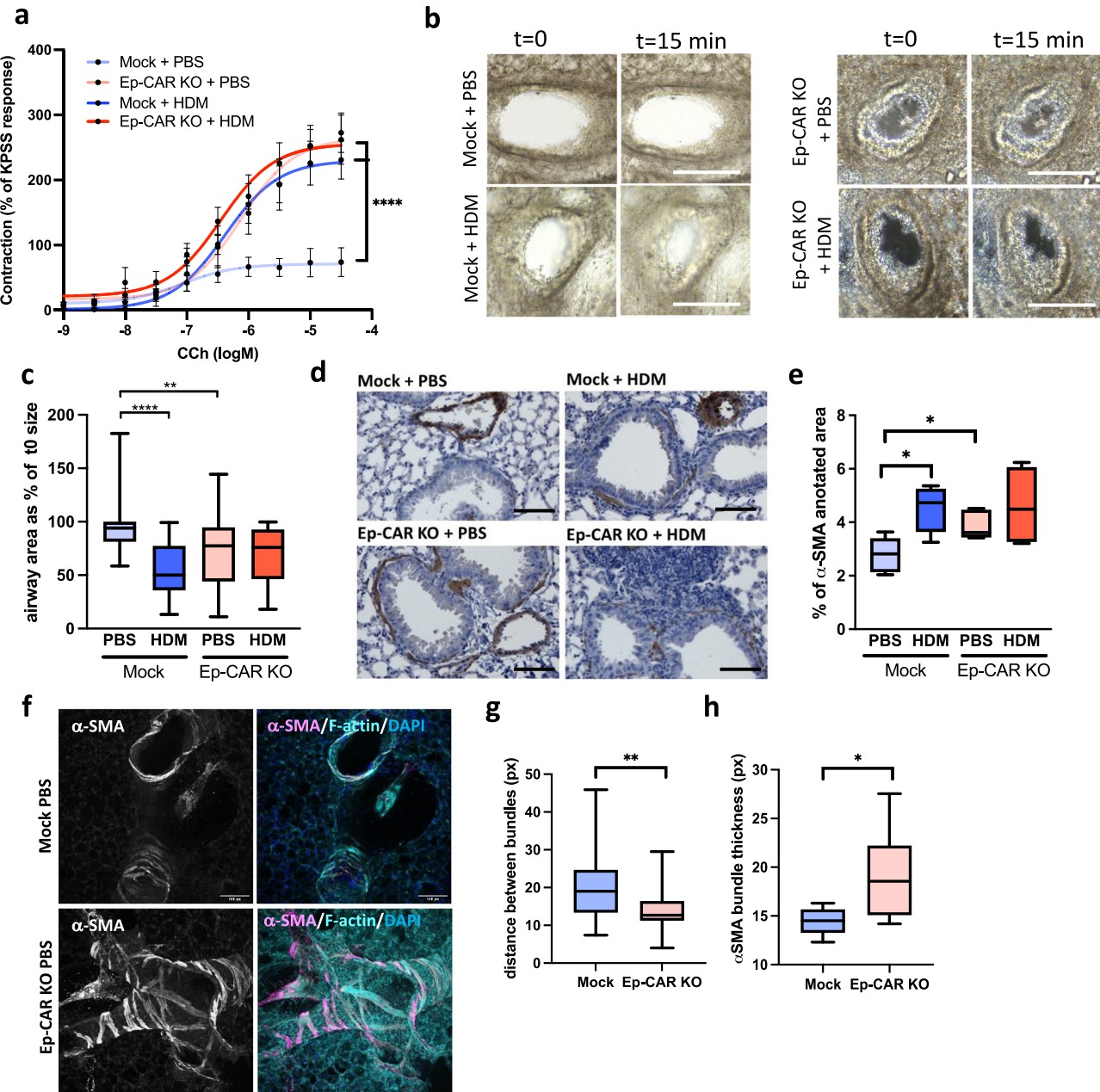

**Fig. 6 | Depletion of epithelial CAR induces airway hypercontractility. a** Airways dissected from Mock and Ep-CAR KO mice treated with PBS or HDM over 5 weeks, mounted on a myograph and treated with carbachol (CCh). Graph shows mean increase in contraction (mN) dose response curves to CCh (1 nM to 30 μM) expressed as a mean % of the high K+ positive control (KPSS). Data shown for 5 airways from 5 mice per group, representative of 2 experiments. **b** Images from PCLS before and after being subjected to 0.5 g/ml of Metacholine for 15 min. Scale bars: 100 mm. **c** Graph showing airway area as % of starting size after 15 min treatment with Metacholine from 25 airways from each experimental group; from 5 mice per condition; representative of 3 independent experiments. **d** Representative images of bronchi from indicated mice stained for α-smooth muscle actin (SMA; brown) and Haematoxylin (nuclei; blue). Scale bars: 50 μm. **e** Quantification of α-

SMA intensity around airways from 25 different airways and 5 mice per condition; representative of 3 independent experiments. **f** Representative confocal Z-projection of PCLS from Mock vs. Ep-CAR KO treated with PBS fixed and stained for a-SMA (magenta), Phalloidin (cyan) and DAPI (blue). Scale bars: 100 mm. **g**, **h** Quantification of distance between F-actin bundles (G) and thickness of α-SMA bundles from images of 30 different airways pooled from 5 mice per condition (1 PCLS/mouse) and representative of 3 independent experiments. All graphs show median (line) with 25/75 percentiles (box) and min/max values except for **a** where mean ± SEM are shown. One-way ANOVA analysis with Dunnett's post hoc test was performed to test statistical significance in **a**, **c**, **e**; unpaired 2-tail student's T-tests were performed on data in **g**, **h**;. $P$ values ** $p < 0.01$; **** $p < 0.0001$. Source data are provided as a Source Data file.

Moreover, supernatants from GSK3β inhibited 16HBE parental cells led to enhanced collagen and fibronectin synthesis by primary lung fibroblasts (Fig. 8h). Thus the activation of the GSK3-β:SMAD2/3 signalling axis leads to enhanced active TGF-β in epithelial cells lacking CAR, which in turns increases the ECM deposition, potentially leading to basal airway hyper-contractility as seen in Ep-CAR KO mice.

## Discussion

We have developed an in vivo model to explore the function of CAR in the lung epithelium and in mediating response to inflammation. Our mouse model combined with in vitro human cell experiments has enabled us to define and dissect signals mediated by CAR, which we summarise in Fig. 9. Under basal physiological conditions, CAR is

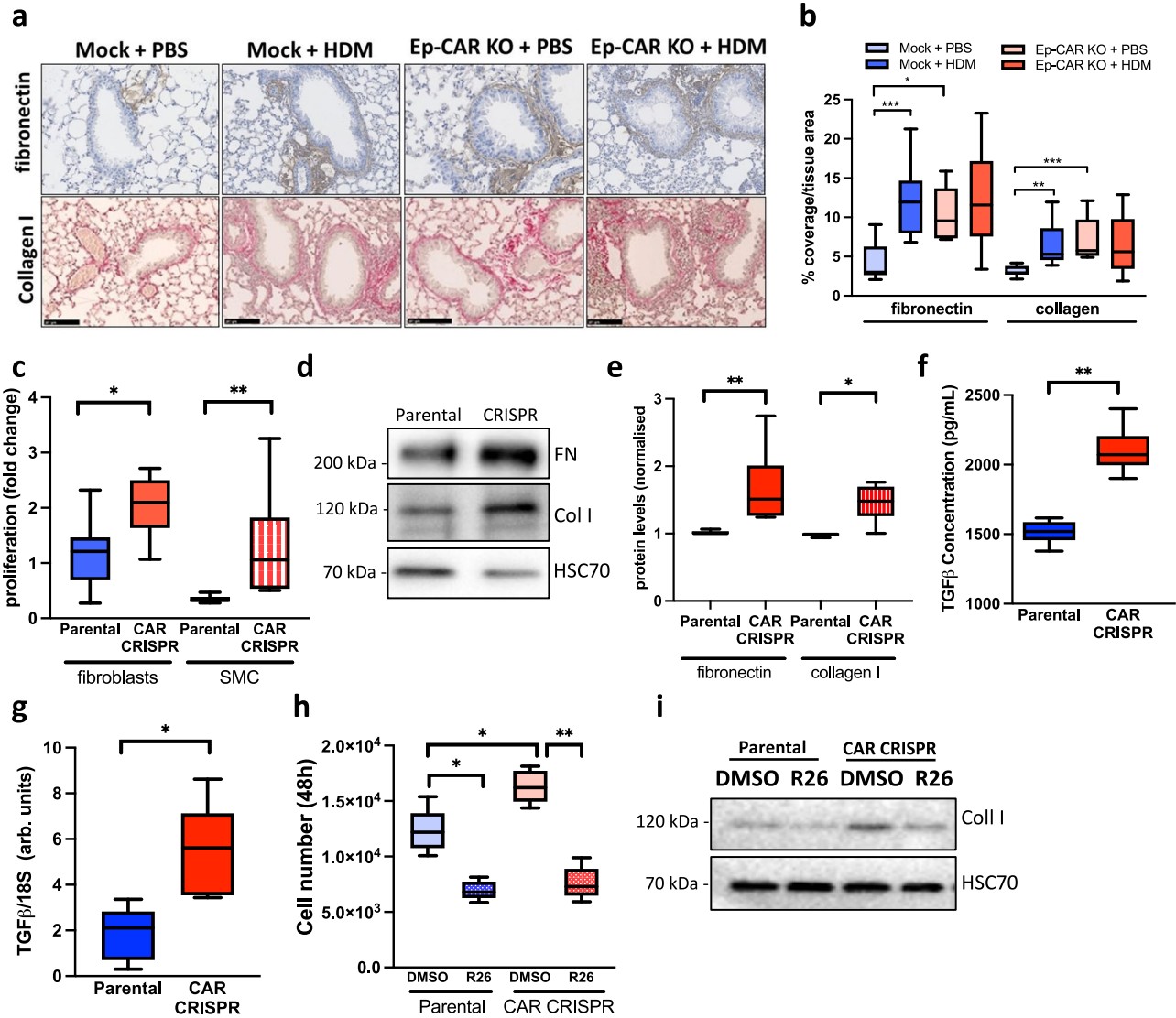

**Fig. 7 | Epithelial depletion of CAR induces TGF-β-dependent airway remodelling. a** Representative images of sections of lungs from mice under specified conditions stained for fibronectin (antibody; brown) and collagen (Sirius red; pink) staining. Scale bar: 100 mm. **b** Quantification of fibronectin and collagen staining around the airways from 20 airways in FFPE lung sections from each experimental group; from 5 mice per condition (one lung per mouse) representative of 3 independent experiments. **c** Quantification of proliferation of primary human lung fibroblasts and airway smooth muscle cells (SMC) treated with supernatants from 16HBE parental or CAR CRISPR cells for 48 h. Data. **d** Lysates from primary human lung fibroblasts treated as in **c** and probed for fibronectin and collagen I. **e** Quantification of 4 independent experiments from data as in **d**. **f** TGF-β levels measured by ELISA in supernatants from 16HBE parental and CAR CRISPR cells. 3 replicates per condition, pooled from 3 independent experiments. **g** TGF-β transcript levels measured by qPCR in 16HBE parental and CAR CRISPR cells. Data shown is pooled from 5 independent experiments. **h** Quantification of proliferation of primary human airway SMC treated with supernatants from 16HBE parental or CAR CRISPR cells for 48 h in the presence or absence of a TGF-β inhibitor (R26). Pooled from 3 independent experiments with 3 replicates per condition per experiment. **i** Lysates from cells as in **h** probed for collagen I and HSC70. Representative of 3 independent experiments. All graphs show median (line) with 25/75 percentiles (box) and min/max values. Unpaired 2-tail student's T-tests were performed on data in **c, e, f, g**; One-way ANOVA analysis with Dunnett's post hoc test was performed to test statistical significance in **b**; two-way ANOVA analysis with Tukey's post hoc test was used for **h**. P values *$p < 0.05$; **$p < 0.01$; ***$p < 0.0005$. Source data are provided as a Source Data file.

engaged in trans at epithelial cell–cell adhesions. Upon challenge with the allergen HDM, CAR is phosphorylated within the cytoplasmic domain, leading to transient destabilisation of CAR and cell–cell adhesions, and triggering upregulation and release of pro-inflammatory cytokines from the epithelium. This leads to enhanced neutrophil and γδ T-cell recruitment to the lung, which drives persistent inflammation and lung matrix remodelling. Depleting CAR from the epithelium prevents HDM-induced inflammation by suppressing HDM-induced cytokine release. However, complete depletion of CAR under non-inflammatory conditions also drives a basal signalling programme, likely due to persistent epithelial barrier destabilisation that promotes matrix deposition and hyper-contractility under basal

conditions. Thus, CAR has a dual function in regulating epithelial architecture and promoting in inflammation in the airways.

Our study focused on HDM as a model for allergic inflammation, as this is the most common risk factor associated with the development of asthma[55]. Interestingly, Ep-CAR KO mice can mount a broadly equivalent response to HDM as seen in control mice, but specific subpopulations of immune cells are not increased in the lung tissue. Our data would strongly suggest that this is due to the significantly reduced levels of key HDM-induced pro-inflammatory factors produced by the lung epithelium in Ep-CAR KO animals following HDM challenge. These factors included IL-4, IL-13, IL-1β and IL-33. This reduction would be expected to reduce lung immune cell infiltration,

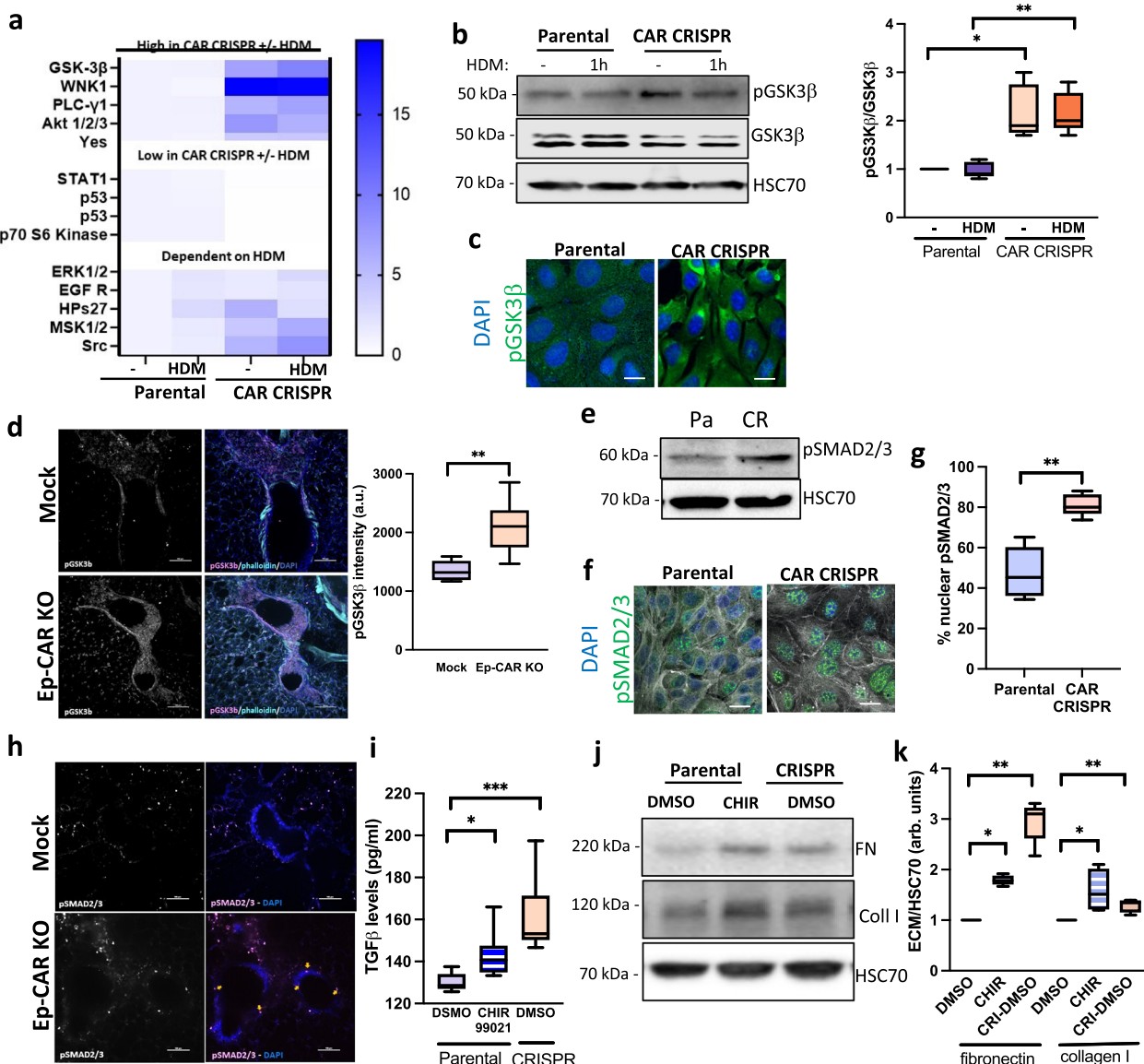

**Fig. 8 | CAR controls a GSK3β-TGF-β signalling axis in lung epithelium. a** Lysates from parental and CAR CRISPR 16HBE cells ± HDM subjected to analysis using the Human Phospho-Kinase Array. Phosphorylation fold changes of target proteins are indicated according to scale bar. **b** Representative western blot of lysates from cells as in **a** probed for pGSK3β, GSK3β and HSC70. Quantification from three independent experiments is shown. **c** Confocal images of parental and CAR CRISPR 16HBE cells stained for pGSK3β (green) and DAPI (blue). Scale bars: 10 mm. **d** Representative confocal Z-projection of PCLS from Mock or Ep-CAR KO mice, fixed and stained for pGSK3b (magenta), F-actin (cyan) and DAPI (blue). Scale bar: 100 mm. Graph to right shows quantification of pGSK3b from 30 different airways pooled from 5 mice per condition (1 PCLS/mouse) and representative of 3 independent experiments. **e** Lysates from parental and CAR CRISPR 16HBE cells probed for pSMAD2/3 and HSC70. **f** Confocal images of parental and CAR CRISPR 16HBE cells stained for pSMAD2/3 (green) and DAPI (blue); F-actin is shown in white.

**g** Quantification of nuclear pSMAD2/3 from images as in **f**. 10 fields of view per condition per experiment quantified, representative of 3 independent experiments. **h** Representative confocal Z-projections of PLCS from Mock and Ep-CAR KO mice stained for pSMAD2/3 (magenta) and DAPI (blue). Scale bar: 100 mm. **i** Lysates from parental and CAR CRISPR 16HBE cells treated with the GSK3β inhibitor CHIR99021 assays for TGF-β levels using ELISA. 3 replicates per condition, pooled from 3 independent experiments. **j** Lysates from cells treated as in **i** probed for fibronectin, collagen I and HSC70. **k** Quantification of western blots as in **j**. Data pooled from three independent experiments is shown in the right. All graphs show median (line) with 25/75 percentiles (box) and min/max values. Unpaired 2-tail student's T-tests were performed on data in **d**, **g**; One-way ANOVA analysis with Dunnett's post hoc test was performed to test statistical significance in **i**; two-way ANOVA analysis with Tukey's post hoc test was used for **b**, **k**. Significance values are *$p < 0.05$; **$p < 0.01$; ***$p < 0.001$. Source data are provided as a Source Data file.

which is indeed what we observe. Interestingly, we also observed upregulation of IL-10 in CAR CRISPR cells in response to HDM. IL-10 has been shown to exhibit immunosuppressive properties in the lung and reduce Th2 responses[56], suggesting loss of CAR may additionally promote factors that actively reduce inflammation. Moreover, as CAR is a direct receptor for molecules expressed on the surface of specific immune cell populations, loss of CAR would also be expected to contribute to reduced tissue integration of immune cells and

interactions with the epithelium. Finally, the increased basal collagen deposition seen in Ep-CAR KO mice may further prevent efficient immune cell tissue infiltration and migration. The combination of these factors very likely explains this lack of tissue-level immune cell infiltration that we report here.

HDM extract can act directly on the epithelium through protease dependent activation of the protease-activated receptor (PAR) family proteins[57], as well as through epidermal growth factor

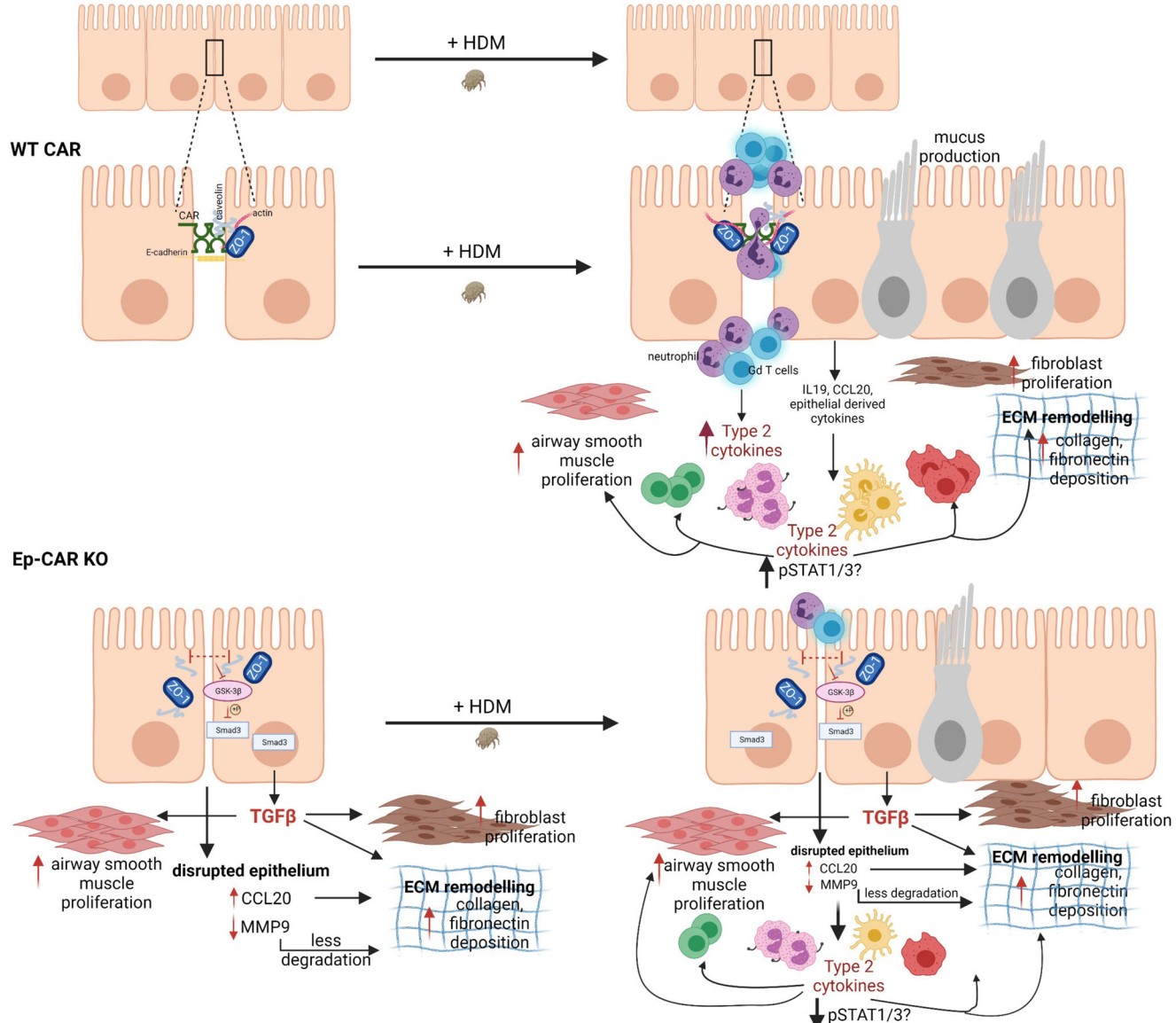

**Fig. 9 | Model of CAR-dependent functions in airway homoeostasis and inflammation.** Top schematic shows proposed functions for CAR in mediating pro-inflammatory factor release to enhance immune cell infiltration following HDM exposure. Bottom panels show destabilisation of epithelial cell–cell adhesions upon loss of CAR leading to enhanced basal airway remodelling and extracellular matrix deposition but reduced HDM-induced pro-inflammatory cytokine release leading to lower levels of immune cell infiltration. See text for further details. Figure created in BioRender.

receptor (EGFR)[58] and Toll-like receptor 4 (TLR4)[59]. In all cases, the resulting signals disrupt epithelial cell adhesion and barrier integrity as we also show here. We demonstrate that HDM induces phosphorylation of CAR and PKCδ activation, potentially as a consequence of induced calcium signalling downstream of these receptors. Importantly we show that blocking CAR homodimers in trans using FK reduces this HDM-dependent phosphorylation, and we have previously shown that FK administration in vivo reduces CAR-dependent inflammation[15], suggesting CAR must be correctly positioned and engaged at the membrane to respond to this signal. We propose that allergen-induced phosphorylation of CAR leads to increased movement of this receptor within the cell–cell adhesion, thereby destabilising the junction whilst simultaneously attracting immune cells through secretion of pro-inflammatory cytokines. Stabilising CAR at the membrane and preventing HDM-induced phosphorylation of CAR would therefore likely significantly reduce HDM-induced inflammation without driving a pro-fibrotic signalling programme seen in the Ep-CAR KO.

Our data place CAR, along with E-Cadherin and ZO-1 as previously reported[60,61], as a central epithelial responder to HDM insult and a key driver of HDM-induced inflammatory signals. Given that CAR is already known to be an important adhesion receptor, it may be difficult to unpick CAR-specific functions versus more general effects on general barrier impairment upon CAR depletion. However, epithelial-specific knockout of E-Cadherin showed similarly impaired epithelial morphology and enhanced mucin levels as we reported in Ep-CAR KO mice, but the E-Cadherin instead showed increased alveolar space and enhanced levels of eosinophils and dendritic cells under basal conditions[62]. This would strongly suggest that the anti-inflammatory effects and enhanced basal matrix remodelling seen in Ep-CAR KO mice are not simply due to epithelial barrier defects, but rather due to CAR-specific effects on the pathways we delineate here. Interestingly, a recent study demonstrated that loss of CAR in breast cancer cells leads to hyperactivation of Akt and GSK3-β kinases leading to TGF-β1-induced epithelial to mesenchymal transition[63]. Whilst the latter study was performed in Ras-transformed mouse mammary tumour cells, the

similarity in signalling phenotype to that described here in bronchial epithelial cells suggest a conserved function for CAR in both lung inflammation and cancer. Data presented here demonstrates that this enhanced TGF-β release from the epithelium in CAR depleted cells correlates with enhanced ECM deposition, and blocking this pathway reduces collagen produced by lung fibroblasts. However, there is also a possibility that the increased TGF-β acts in an autocrine fashion to promote epithelial destabilisation in an 'EMT-like' fashion. Future experiments targeting this signalling axis in vivo in Ep-CAR KO mice will enable determination of the relative contributions to epithelial vs. lung stromal cell behaviour. One key distinct feature of CAR is in the ability of this receptor to bind to proteins on the surface of neutrophils and γδ T cells, and in doing so contributes to transepithelial migration of immune cells to the inflammation site. The contribution of CAR to immune cell infiltration is also evident in Ep-CAR KO mice, where we show dysregulated secretion of key Th2 cytokines that affect the profile of immune cell infiltration. Whilst HDM challenge of E-Cadherin lung epithelial knockout mice have yet to be investigated, evidence to date would suggest that cell–cell adhesion molecules in the epithelium play distinct and diverse functions in mediating lung homoeostasis.

Our study also showed several further proteins that can complex with CAR in lung epithelial cells. We validated caveolin-1 as a CAR interactor, and whilst it remains unclear if this is mediated by direct or indirect binding, our data strongly suggest this complex is sensitive to phosphorylation of CAR. We postulate that this complex serves to scaffold other molecules with CAR at the plasma membrane and aides in the translation of mechanical signals from the outside to signalling pathways inside the cell. Interestingly, caveolin-1 has been reported to be important for maintaining E-cadherin levels and barrier function in human lung epithelial cells. Caveolin-1 membrane localisation is disrupted by HDM challenge (the latter of which we also show in our study), leading to increased levels of TSLP, a pro-allergic factor involved in Th2 responses in asthma[38]. Moreover, caveolin-1 knockout mice show progressive increase in TGF-β-dependent deposition of collagen fibrils in airways and parenchyma[64], similar to the phenotypes we report in Ep-CAR KO mice here, and further supporting the notion of co-operativity between CAR and caveolin-1 in mediating response to inflammation. Notably we also identified the mechano-responsive transcription factor YAP1 as a CAR-associated molecule. YAP is a key factor in lung development, and of relevance to our study, has also been shown to control TGF-β activation in the lung epithelium[65]. Moreover, a recent report of a lung epithelial-specific knockout of YAP demonstrated disrupted epithelial barrier formation and mucus hypersecretion[66], mirroring phenotypes seen in our Ep-CAR KO mice. Combined, these findings place CAR as a pivotal membrane-bound receptor in scaffolding key players involved in epithelial environmental sensing and response. In addition to caveolin-1 and YAP, we also identified several integrin receptor subunits complexed with CAR, which may be important in the ECM adhesion defects observed in CAR CRISPR cells. Indeed, we have recently demonstrated that CAR can complex with integrins to control their activation status in the context of cancer[67]. Endocytic proteins, including VAMP-3, sorting nexin-9 and Rab23 were also enriched in complex with CAR; these would be interesting to investigate in future as potential mediators of CAR traffic and subcellular positioning under homoeostatic and inflamed conditions.

In summary, our study demonstrates additional functions for CAR in maintenance of lung homoeostasis and potentiation of allergic-induced inflammatory response. Whilst removal of CAR altogether appears to drive a pathological response, we propose that acutely manipulating CAR function, suppressing HDM-induced phosphorylation and stabilising this receptor at the membrane might present a new opportunity to target inflammation in chronic lung disease. This will be an interesting possibility to explore in future.

## Methods

All the research in this manuscript complies with all ethical regulations. The use of animals for this study was approved by the Ethical Review Committee at King's College London and the Home Office, UK. All animals were housed in the Biological Support Unit (BSU) located in New Hunt's House at King's College London. All experiments were carried out under project license number P9672569A and personal license number I0F9CA46A.

### Reagents and antibodies

The following antibodies were used: Anti-CAR (H300; 1:200) was from Santa Cruz Technology. p-CAR Thr$^{290}$/Ser$^{293}$ polyclonal antibody was previously described (14) and was developed by Perbioscience (Thermofisher) using the peptide Ac- RTS (pT)AR(pS)YIGSNH-C and was affinity purified before use. Anti-phospho-ERK (T202/Y204; 1:1000), anti-ERK (1:500), anti-phospho-PKCδ (1:500) and anti-phospho STAT1 (1:1000) antibodies were from Cell Signalling. Anti-caveolin-1 (1:500), anti-collagen I (1:500), anti α-SMA (1:1000), anti-LyC6G (1:400), anti-E-cadherin (1:500) and anti-IL19 (1:250) antibodies were from Abcam. Anti-fibronectin (1:1000), anti-Club (1:250) antibody anti-HSC70 (1:2000) antibodies were from Sigma. Anti-ZO-1 (1:500) antibody was from Millipore. Anti-GFP (1:1000) antibody was from Roche. Anti-phospho-Ser9 GSK3β (1:500), anti-GSK3β (1:400) and anti-phospho SMAD2/3 (1:500) were from R&D Systems. Antimouse HRP (1:2000) and anti-rabbit-HRP (1:2000) were from DAKO. Antimouse-568 (1:400), antimouse-488 (1:400), anti-rabbit-568 (1:400) and phalloidin-647 (1:500) were all obtained from Invitrogen. CalyculinA, sodium orthovanadate and protease inhibitor cocktail 1 were obtained from Calbiochem. CAR and non-targeting siRNA were transfected using Dharmafect1 (Dharmacon). For Flow Cytometry analysis, we used the following antibodies: γδ TCR- Alexa421, CD45-PerCP Cy5, CD3-FITC, CD4-PE, B220 PECy7, CD8 APC Alexa657, NK1.1 APC-Cy7, Lyc6G Alexa 421, CD11c FITC, CD11b PE and Siglec F-Alexa700. All the antibodies were used at 1:50 and purchased from BioLegend except for Siglec F-Alexa700 which was from BD Biosciences (1:50). CHIR99021 and R268712 were from BioTechne. Calyculin A was obtained from Calbiochem. Adenovirus Type5 Fiberknob (FK) was produced and purified as previously described[33]. House Dust Mite extract (HDM) containing *Dermatophagoides pteromyssimus* was from Citeq biologics (02.01.85).

### Plasmids

Full length and mutant CAR lentiviral plasmids were described previously[68]. Phospho-mutant CAR constructs were generated using site directed mutagenesis and have been described previously[14,15]. siCAR and a non-targeting control siRNA pool were from Dharmacon. For the Bio-ID experiments in 16HBE CAR CRISPR cells, a full-length version of CAR with a TurboID tag[69] inserted between the PDZ-binding-domain and the transmembrane domain was cloned into a pcDNA3.1 backbone. Constructs were generated by PCR-based gene assembly to integrate the TurboID[2] between the transmembrane domain and the cytoplasmic tail. This strategy was chosen to retain the free C-terminal S/T-X-V motif that is required for interaction with PDZ-domains. The tag was inserted between Lys285 and Ser286 after a stretch of prolines to reduce the chance of interference with any essential structural feature. Localisation to the plasma membrane and restoration of cell–cell adhesion in CAR CRISPR cells was verified indicating proper maturation and trafficking of the CAR-TurboID fusion protein. CAR CRISPR plasmids were in a pSpCas9 GFP vector from GeneScript. Two different sequences/plasmids were used as follows: CXADR CRISPR guide RNA1: ACGTAACATCTCGCACCTGA and CXADR CRISPR guide RNA 2: AGTACCTGCTAACCATGAAG. CRISPR cells were expressing GFP transiently were enriched by FACS sorting, followed by bulk culture and verification of CAR deletion by qPCR, Western blotting and sequencing.

## Cell culture and transfections

All cell lines were maintained under standard conditions (37 °C, 5% $CO_2$ and 95% humidity). 16HBE human bronchial epithelial cells were a gift from Prof D. Gruenert (University of Vermont, US;[70]) and grown on type I collagen coated tissue culture flasks. Cells were cultured in Modified Eagle's Medium (MEM supplemented with 10% Fetal Bovine Serum (FBS) (Gibco/ThermoFisher), 1% glutamine and 1% penicillin/streptomycin (Sigma-Aldrich). HEK 293T cells (used for lentiviral production) were purchased from ATTC and cultured in DMEM supplemented with 10% FCS (Gibco/ThermoFisher), 1% glutamine and 1% penicillin/streptomycin (Sigma-Aldrich). Primary human lung fibroblasts were a gift from Prof Jenkins (National Heart and Lung Institute, Imperial College London, UK), and were cultured in DMEM supplemented with 10% FCS (Gibco/ThermoFisher), 1% glutamine and 1% penicillin/streptomycin (Sigma-Aldrich). Airway smooth muscle cells were a gift from Dr. Woczcek (King's College London, UK), and were grown from bronchial biopsies by explant culture as previously described[71]. In all cases, cells were subculture using 0.05% trypsin-EDTA in PBS. Transfections in 16HBE cells were performed using lipofectamine3000 (Thermo-Fisher) following manufacturer's instructions. 16HBE CAR CRISPR cells re-expressing CAR-GFP were generated using lentiviral transduction of WTCAR-GFP as described previously[14]. All cells were routinely tested for mycoplasma every month and were confirmed negative.

## RNA isolation and reverse transcription polymerase chain reaction (RT-PCR)

RNA was isolated from cells or lung tissues using the RNeasy Qiagen kit (Qiagen; cat no 74004) following the manufacturer's instructions. In the case of isolating RNA from lung tissue, lung lobes were chopped and stores in Trizol. The tissue was then disaggregated using a Bullet Blender. RNA was used for cDNA synthesis using the LunaScript RT Super-Mix Kit (NEB; cat. no. E3010). cDNA synthesis reaction, consisting of 4 μL of 5X LunaScript RT SuperMix, 1 μg of RNA and Nuclease-free water, was prepared. Using a thermal cycler, the reaction was initialised by a primer annealing step of 25 °C for 2 min, followed by a cDNA synthesis step of 55 °C for 10 min, and a heat inactivation step of 95 °C for 1 min. Quantitative real-time PCR (qPCR) was performed using a QuantStudio 5 (Applied Biosystems/ThermoFisher) thermal cycler, the reaction was first heated to 95 °C for 1 min, followed by 40 cycles of 95 °C for 10 s. This was followed by an extension time of 30 s at 60 °C. The mouse probes were used as follows: *Il4* (Mm00445259_m1), *Il13* (Mm00434204_m1), *Il1b* (Mm00434228_m1), *Ccl20* (Mm01268754_m1), *Cxadr* (Mm00438355_m1). The following human probes were used: *MMP9* (Hs00957562_m1), *MMP10* (Hs00233987_m1), *MMP13* (Hs00942584_m1), *LIF1* (Sc04122526_s1), *Il19* (Hs00604657_m1), *TGFβ1* (Hs00998133_m1) and *18S* (Hs03003631_g1).

## GFP-TRAP Immunoprecipitation

16HBE cells expressing GFP-tagged proteins were cultured on 10 cm dishes until 100% confluent. Cells were then washed three times with ice-cold PBS before 500 μl of cold GFP-trap buffer (50 mM Tris (pH 7.4), 150 mM NaCl, 1% NP40, 15 mM $MgCl_2$, 10% glycerol, 1% Triton X-100, 5 mM EDTA containing protease and phosphatase inhibitor cocktails (1:100) was added. Cells were subsequently scraped and centrifuged at 5000 × g for 10 min at 4 °C. 1:1 of GFP-trap beads and control agarose resin was washed three times with IP lysis buffer. Supernatants obtained from cell lysates were added to the beads and incubated on a rotator at 4 °C for 2 h. 50 μl of supernatants were also set aside as input. Afterwards, the beads were washed three times with lysis buffer. 2× sample buffer containing β-mercaptoethanol (1:100) was added to the beads, boiled for 5 min at 95 °C and centrifuged to clear cell debris. 40 μl of samples were loaded and then subjected to Western blot analysis.

## LC-MS/MS analyses of BioID pulldown samples

Identification of CAR protein interactors in 16HBE cells were performed as in[72]. For each condition 8 ×15-cm-cell culture dishes were transfected using Lipofectamine 3000 (Thermofisher). 2 days after transfection cells were incubated with 500 μM Biotin in medium for 15 min, then washed 3 times with PBS and harvested immediately. For each replicate, cells from 2 dishes were pooled, pelleted and snap frozen for subsequent pulldown. Cell pellets were resuspended with 700 μl of BioID lysis buffer (50 mM Tris pH 7.5; 500 mM NaCl; 0.4% SDS; 5 mM EDTA; 1 mM DTT; 2% Triton X and 1× PICS I). The samples were incubated for 60 min on ice and tubes were mixed in between followed by sonication. To remove insoluble material the samples were centrifuged for 15 min at 4000 × g at 8 °C. The supernatant was transferred into a new tube and incubated with the beads (MyoOne Streptavidin C1; Invitrogen) overnight. Beads were collected with a magnetic rack and washed 2 times for 8 min with 0.1% deoxycholate; 1% NP-40; 500 mM NaCl; 1 mM EDTA and 50 mM HEPES pH 7.5 at 4 °C, 2 times washed with 250 mM LiCl, 0.5% NP-40, 0.5% deoxycholate, 1 mM EDTA, and 10 mM Tris pH 8.1 at 4 °C, and 2 times washed with 50 mM Tris pH 7.4 and 50 mM NaCl at 4 °C and dried using a speedvac before resuspension. Beads were resuspended in 20 μl urea buffer (6 M urea, 2 M thiourea, 10 mM HEPES pH 8.0), reduced in 10 mM DTT solution, followed by alkylation using 40 mM chloroacetamide. Samples were first digested with 1 μg endopeptidase LysC (Wako, Osaka, Japan) for 4 h and, after adding 80 μl 50 mM ammonium bicarbonate (pH 8.5), digested with 1 μg sequence-grade trypsin (Promega) overnight. The supernatant was collected and combined with the supernatant from an additional bead wash in 50 mM ammonium bicarbonate (pH 8.5). Samples were acidified with formic acid and peptides were desalted using C18 columns.

Peptides were eluted from C18 columns, dried using a speedvac, resuspended in 3% acetonitrile/0.1% formic acid and separated on a 20 cm reversed-phase column (ReproSil-Pur 1.9 μm C18-AQ resin, Dr. Maisch GmbH) using a 98 min gradient with a 250 nl/min flow rate of increasing acetonitrile concentration (from 2% to 60%) on a high-performance liquid chromatography system (Thermo Scientific). Peptides were measured on a Q Exactive Plus instrument (Thermo-Fisher Scientific), operated in the data-dependent mode with a full scan in the Orbitrap (70 K resolution; $3 \times 10^6$ ion count target; maximum injection time 50 ms) followed by top 10 MS2 scans using higher-energy collision dissociation (17.5 K resolution, $5 \times 10^4$ ion count target; 1.6 m/z isolation window; maximum injection time: 250 ms).

Raw data were processed using MaxQuant software package (v1.6.3.4). The internal Andromeda search engine was used to search MS2 spectra against a decoy human UniProt database (HUMAN.2019-07) containing forward and reverse sequences, including the sequence for the CAR-BirA construct. The search included variable modifications of methionine oxidation and N-terminal acetylation, deamidation (N and Q), biotin (K) and fixed modification of carbamidomethyl cysteine. Minimal peptide length was set to seven amino acids and a maximum of two missed cleavages was allowed. The false discovery rate (FDR) was set to 1% for peptide and protein, and site identifications. Unique and razor peptides were considered for quantification. MS2 identifications were transferred between runs with the 'match between runs' function and IBAQ intensities were calculated using the built-in algorithm. The resulting proteinGroups text file was filtered to exclude reverse database hits, potential contaminants, and proteins only identified by site. Statistical data analysis was performed using Perseus software (v1.6.2.1). Log2 transformed IBAQ values were filtered for minimum of 3 valid values in at least one group and missing values were imputed with random low intensity values taken from a normal distribution. Differences in protein abundance between CAR-BirA samples and control samples were calculated using two-sample Student's *t*-test. Proteins enriched in the CAR-BirA group and passing the significance cut-off (permutation-based FDR < 5%, minimum 3

peptides identified, minimum 4 MS/MS counts) were considered CAR-BirA interactors. GO term enrichment analyses for CAR-associated proteins were performed using Cytoscape and the ClueGO plugin.

The mass spectrometry proteomics data have been deposited to the ProteomeXchange Consortium via the PRIDE partner repository with the dataset identifier PXD029237

## Western blotting
Cells were lysed in buffer (0.05 M Tris-HCl, 0.15 M NaCl, 1% Triton X-100, pH 7.2), containing protease and phosphatase inhibitors. After centrifugation, proteins in the supernatant were quantified, boiled with Laemmli buffer, resolved by SDS-PAGE and transferred to a nitrocellulose membrane. Western blotting was performed using standard procedures. Proteins were then detected with ECL chemiluminescence kit (Bio-Rad Laboratories; cat.no. 1705061) and imaged (ChemiDoc Imaging Systems, Bio-Rad Laboratories). Blots were analysed and processed using Image Lab (v5.2.1, Bio-Rad Laboratories).

## Cell adhesion assays
Collagen (Rat Tail Type I, Corning; 50 μg/ml) or Matrigel (BD Biosciences, 1:2 dilution) was incubated in a 24-well plate and allowed to coat the surface for 1 h at 37 °C. $1\times10^3$ cells were seeded in each well. Cells were allowed to adhere for different times (30 min to 2 h) at 37 °C. Cells were then washed with PBS, before fixing with 4% paraformaldehyde (PFA/PBS) for 10 min. After washing with PBS, cells were stained via incubation with DAPI and Alexa-Fluor 488 Phalloidin for 30 min. Cells were finally washed with PBS. Fluorescent images were acquired on an Evos FL Auto 2 fluorescent microscope (Thermofisher). Tile scans were obtained using a 4× air objective using identical camera acquisition times. Tiles were knit into.TIFF files using FIJI software and total cell count was obtained by thresholding for nuclear stain followed by automated counting.

## Cell proliferation assay
Human Derived Fibroblasts (HDF) cells or airway smooth muscle (ASM) cells were plated on a 24-well plate and incubated at 37 °C 5% $CO_2$. The next day, cells were incubated with different supernatants (16HBE, 16HBE CAR CRISPR with or without treatment), and where incubated for 24 or 48 h post-plating. Cells were then washed with PBS, before fixing with 4% PFA/PBS for 10 min. Nuclei were stained with DAPI to enable cell quantification. Cells were imaged on an Evos FL Auto 2 fluorescent microscope (Thermofisher). 9×9 tile scans were obtained using a 10x air objective with 3.2 MP CMOS camera. Ex$^{cit}$ation using DAPI LED light cube was used. Images were acquired using EVOS software (v2). Tiles were knit into.TIFF files using FIJI software and total cell count was obtained by thresholding for nuclear stain followed by automated counting. Using three biological repeats per condition, an average number of cells per condition was calculated.

## Transepithelial migration experiments
16HBE parental, 16HBE CAR CRISPR or 16HBE CAR CRISPR + CAR-GFP (rescued) cells were plated in 24-well plate. After 24 h, HDM was added for 24 h and the supernatant was used to feed the lower chambers as follows. HL60 cells were seeded in 6.5 mm Transwell chambers (Corning, UK) with 8.0 μm pores at 30,000 cells per chamber. Media in the lower chambers was changed to supernatants from 16HBE cells treated or not with HDM as explained above. HL60 cells were allowed to migrate for 6 h. Cells that migrated through the Transwell in 6 h were stained with Cell Tracker Orange Dye (Molecular probes, UK) and fluorescent images were acquired on an Evos FL Auto 2 fluorescent microscope (Invitrogen). Tile scans were obtained using a 4x air objective using identical camera acquisition times. Tiles were knit into.TIFF files using FIJI software and total cell count was obtained by thresholding for nuclear stain followed by automated counting.

## ELISA assay
Sandwich ELISA was used to detect supernatant analytes. Cell-free supernatants were extracted from wells 24 h after cell monolayers had formed and kept at −20 °C until analysis. ELISA kits for IL-4 (D4050), IL-19 (D1900) and TGF-β (Dy240-05) were used and were all from R&D Systems. In all cases, protocols were conducted according to the manufacturer's instructions and detected on a Victor 1420 multilabel counter (Perkin Elmer) quantifying concentrations drawn from a standard curve on each plate.

## Dextran permeability assay
16HBE cells (parental or CAR CRISPR or rescued with CAR GFP) were plated at 10000 cells per well in 0.2 ml normal growth media in the upper chamber of 0.4 μm pore size Transwell pre coated with collagen, with 0.6 ml of growth media in the lower well. After 24 h, wells were checked for the formation of complete cell monolayers. At this point media was removed from the upper and lower wells and replaced with fresh growth media with or without 5 mM EDTA and incubated for 30 min at 37 °C. 10 μl of TRITC-dextran (20 KDa) solution was then added to the upper chamber of all wells and placed back in the incubator. 100 μl of media was collected from the lower chamber after 2 h on incubation. The media fluorescence was then measure using a fluorescence plate reader.

## Gelatin degradation assay
To analyse gelatin degradation, the Abcam kit: Gelatin Degradation Assay Kit (Alternative to Zymography) was used, following manufacturer's instructions. Briefly, 16HBE cells (parental or CAR CRISPR) were plated ($1\times10^6$ cells/condition) and serum starved. 24 h later, HDM (2.5 μg/mL) or PBS was added for 24 h. Cells were then lysate with 100 μL of the Cell Lysis Buffer provided by the kit. A standard curve was prepared, and then gelatinase substrate mix was added to each sample and to the positive control well. Fluorescence at ex/em 490/520 nm was measured in kinetic mode at 37 °C for 1–2 h.

## Cytokine and Phospho-kinase antibody arrays
Cells were treated with HDM (2.5 μg/mL) or PBS as a control for 24 h before being subjected to a Proteome Profiler Human XL Cytokine Array analysis (R&D Systems, Inc.) as per manufacturer's instructions. For Kinase Array analysis, 16HBE cells or 16HBE CAR CRISPR cells were plated overnight. The next day, cells were serum starved overnight and then treated with HDM (2.5 μg/mL) or PBS as a control for 24 h before being subjected to Human Phospho-Kinase Antibody Array (R&D Systems; ARY003B) as per manufacturer's instructions. Data for the latter assay is presented with each row (representing a different cytokine/molecule) scaled to show the smallest (min, dark blue) and maximum (max, red) densitometry readings for that molecule, meaning each row is scaled independently of the others in terms of absolute values. The data is presented in this manner due to the very large differential in the levels of each molecule represented on the array and provides means to evaluate and display the relative changes in expression levels of each target across the different conditions

## Immunofluorescence and confocal microscopy
Cells on coated coverslips were plated, and 24 h later there were washed with PBS, fixed with 4% PFA in PBS for 10 min and permeabilised with 0.2% TritonX-100 for 10 min. Cells were incubated with primary antibodies overnight and appropriate secondary antibodies and Phalloidin conjugated to Alexafluor 568 or 633 for 1 h. Cells were mounted onto slides using FluorSave (Millipore). Confocal microscopy was performed using a Nikon A1R inverted confocal laser scanning microscope with a 60x oil objective and laser excitation wavelength of 488 nm (for GFP or Alexafluor-488), 561 nm (for Alexafluor-568) and 633 nm (for Alexafluor-633 and cy5). Images were exported from the

Nikon Elements software (Nikon) for further analysis in ImageJ software (U. S. National Institutes of Health, Bethesda, MD, USA).

For live cell imaging experiments, cells were plated onto glass bottomed Ibidi chambers (Ibidi, Germany) and allowed to form an intact monolayer. Cells were then imaged every minute using 488 nm laser excitation using a 60x oil objective on a Nikon A1R inverted confocal microscope (Nikon UK) equipped with a humidified environmental chamber heated to 37 °C, with 5% $CO_2$ and PFS activated. HDM was added to the imaging media at a final concentration of 2.5 µg/mL and imaging immediately resumed every minute for a further 60 min. All images were saved as.nd2 files and analysed in NIS Elements software (Nikon) or exported as tif files for presentation. Analysis of live cell junctional dynamics was performed using ImageJ. Individual cell–cell junctions were randomly selected from the final frames of movies using a defined ROI (90 in total per condition and cell type). Images were acquired as described above. In ImageJ, lines of 20 µm in length and 1 µm in width were drawn perpendicular to junctions identified using the specified protein in the relevant figures. Lines were aligned so that the 10 µm point was at the centre of junctions. The intensity values of each channel were exported and the individual data values for the specified proteins for each cell were normalised to control cell values. The resultant graphs show a peak of intensity where the junctional protein forms a sharp, defined line at the cell–cell contact point. Flatter traces without peaks indicate loss or dispersion of the protein of interest from the junction.

## Generation of CAR-deficient mice
Scgb1a1-CreER were purchased from The Jackson Laboratory. These mice express a tamoxifen-inducible form of cre recombinase from the *Scgb1a1* locus (secretoglobin), inducing cre recombinase activity in bronchiolar non-ciliated Club cells. These mice were crossed with mice carrying floxed alleles for *CAR* (B6;129S2-Cxadrtm1.1Ics) (19). C57BL/6 mice (strain 027) were purchased from Jackson Laboratories. Intraperitoneal administration of Tamoxifen (Sigma) was used to induce the lung epithelial expression of Cre and excision of *CAR*. Tamoxifen was administered using 75 mg tamoxifen/kg body mouse weight for 6 consecutive days and lungs were extracted after 2 weeks from the end of the Tamoxifen treatment for validation. Tamoxifen was diluted in corn oil (Sigma). Control mice (mock, no KO generated) received corn oil intraperitoneally. All mice were used between 6 and 12 weeks of age.

## House Dust Mite (HDM) sensitization
For House Dust Mite (HDM) immune-sensitised protocol, 6-8 weeks female C57BL/6 mice, mock or lung Ep-CAR KO) were anaesthetised with isofluorane and administered either 25 µg (total protein) of HDM extract (Citeq Biologics; 1 mg/ml protein weight solution dissolved in PBS) or 25 µl of PBS intranasally 5 times per week for 5 consecutive weeks. Control mice received 25 µl of PBS. Mice were culled 24 h after the final HDM or PBS dose. Where relevant, HDM was first dosed to animals 3 weeks following the initial Tamoxifen injection. We note that we performed initial characterisation experiments in both male and female C57BL/6 control mice and found higher variability in the males in terms of immune cell populations in the BAL and tissue following HDM challenge compared to females. Whilst we are unsure as to the reason for this (and have not seen it reported specifically in the literature), we felt unknown confounding factors from the use of mixed sex populations may make interpretation of data challenging, so we chose to perform all subsequent experiments in female mice.

## Cell recovery and flow cytometry
Bronchoalveolar lavage (BAL) was performed by washing the airways three times with 400 µl of PBS, after centrifugation supernatants were stored at −80 °C for further analysis and cells resuspended in 500 µl of complete media (RPMI, 10% FCS, 2 mM L-glutamine, 100 U/ml penicillin/streptomycin) (GIBCO, Life Technologies). The left and inferior lung lobes were chopped and digested in complete media supplemented with 0.15 mg/ml collagenase (Type D; Roche) and 25 µg/ml DNase (Type 1; Roche) for 1 h at 37 °C. Tissue was then passed through a 70 µm sieve (BD Bioscience), washed, and resuspended in 1 ml of complete media. in 1 ml of complete media. Red blood cells in 200 µl whole blood were lysed and the remaining leucocytes were washed twice and then resuspended in 1 ml of complete media. Cells were washed and incubated for 20 min with rabbit serum (Sigma-Aldrich) prior to staining for extracellular antigens in 5% FCS / 1% BSA in PBS for 30 min at 4 °C. Cells were washed, fixed, and permeabilized using Fix/Perm kit (eBioscience; 88-8824-00) before being stained for intracellular antigens. Detection was done using Attune NxT flow cytometer (ThermoFisher Scientific) and the resulting data were analysed using FlowJo software (Tree Star, Ashland, Ore). Flow cytometry gating conditions are provided in Supplementary Fig. 7.

## Wire myography of ex vivo mouse airways
Bronchi were dissected from the left lung, and the right superior and inferior lobes of mice treated with corn oil (Mock), tamoxifen (Ep-CAR KO), and challenged or not with house dust mite (HDM) for 5 weeks. Isolated bronchi were mounted on force transducers using 30 µm wire on a Multi Myograph Model 610 M (Danish Myo Technology) in 5 ml organ baths containing Physiological Salt Solution (PSS; 118 mM NaCl; 4.7 mM KCl; 1.2 mM KH2PO4; 1.2 mM MgSO4; 25 mM NaHCO3; 2.5 mM CaCl2; 0.003 mM ETDA.Na2.2H2O; 11 mM glucose all Sigma-Aldrich) perfused with carboxygen (oxygen plus 5% $CO_2$) and stretched to an appropriate resting tension (1–2 mN). High K + Physiological Salt Solution (KPSS; isotonic replacement of NaCl by KCl) was used as a positive control. Cumulative concentration responses to carbachol (Sigma-Aldrich) ($10^{-9}$–$10^{-6}$ M at 0.5 log10 intervals) were then performed. Increase in contraction tension (mN) in response to each concentration was measured and data normalised to KPSS responses.

## Ex vivo precision cut lung slices
Ex vivo lung slices were obtained from mice, within 24 h of the last PBS or HDM dose, adapted from the previously published protocol[73]. Briefly, mice were humanely killed by $CO_2$ inhalation followed by cervical dislocation. The chest cavity was opened and the trachea carefully exposed, where a small incision was made to accommodate the insertion of a 20Gx1.25 in canula (SURFLO I.V. catheter). The lungs were inflated with 2% low melting agarose (Fisher) prepared in HBSS (Gibco) before lungs, along with the heart and trachea, were excised, washed in PBS, and the lobes separated. Individual lobes were then embedded in 4% low melting agarose and solidified on ice. 200µm thick slices were cut on a Leica VT1200S vibratome and washed and incubated in DMEM/F-12 medium supplemented with 10% fetal bovine serum (FBS) and antibiotics. The PCLS remained viable (and MCH-reactive) for at least 12 days after isolation. Lung slices were prepared and if used for fixed immunofluorescence, were fixed in 4% PFA on the same day as preparation.

## Precision cut lung slice imaging
PFA fixed ex vivo lung slices were incubated for 1 h at room temperature in blocking solution: PBS containing 0.1% triton X-100, 0.1% sodium azide, and 2% bovine albumin (BSA) before incubating overnight a 4 °C in 1:100 rabbit anti-E-Cadherin (Cell Signalling) in blocking solution. Ex vivo lung slices were washed in PBS before incubating overnight with Alexa-Fluor 488 goat ant-rabbit IgG (Thermo Scientific; 1:500) and Alexa-Fluor 568 Phalloidin (Thermo Scientific; 1:500) and DAPI, washed in PBS mounted, and imaged on a Nikon Eclipse Ti2 spinning disc confocal microscope with a 20X objective.

For live imaging, PCLS were placed overnight in growth medium, used 24 h post-isolation and treated with increasing doses of MCH (acetyl-β-methylcholine chloride, Sigma), from 100 mg/mL to 500 mg/mL in HBSS+ solution. PCLS were incubated in HBSS+ in 24-well plates at 37 °C with MCH for 30 min, imaged at 30 s intervals using a Life

Technologies EVOS FL Auto microscope to measure bronchoconstriction in response to MCH. PCLS were fixed in 4% paraformaldehyde overnight at 4 °C before immunostaining. PCLS were viable and Methacholine-responsive for up to 12 days post-isolation.

## Tissue processing and analysis

Lungs were extracted from mice post-sacrifice, were fixed with 4% paraformaldehyde for 24 h post-excision and subsequently moved into 70% Ethanol prior to paraffin wax embedding. 10 μm thick sections were obtained from each paraffin-embedded or frozen specimen. Tissue sections were then used for different staining, including Haematoxylin and Eosin or Periodic Acid-Schiff (PAS) staining to visualise Goblet cells. Immunohistochemistry was also perform using DAB staining. Paraffin-embedded sections were melted at 95 °C for 2 h and de-waxed by dipping slides in xylene 2 × 10 min, 100% EtOh 2 × 5 min, 70% EtOh 1x5min and 50% EtOh 1x5min. Antigen retrieval was carried using sodium citrate buffer (0.0874 M sodium <sup>cit</sup>rate, 0.0126 M citric acid, pH 6) and incubating the slides for 20 min in a pressure cooker at 95 °C. Endogenous peroxidase activity was blocked via 10 min incubation in hydrogen peroxide (3% in TBS) for DAB staining. Tissues were then washed 3× with TBS and non-specific binding was blocked via incubation with TBS-1%BSA-1%FBS blocking solution for 1 h at room temperature. Primary antibody was added to the tissues, and these were left at 4 °C overnight. After 3× washes with TBS, tissues were incubated with fluorescent or HRP-conjugated secondary antibodies for 1 h at room temperature. Tissues used for fluorescence were mounted with Fluorsave solution (Millipore). DAB staining was visualised by adding DAB developing solution for up to 20 min (Dako). Tissues were then counterstained using haematoxylin for 1 s. Finally, tissues were dehydrated with graded alcohols and xylene before being mounted with DPX.

Lung sections were imaged using a pathology slide scanner (Hamamatsu) and analysed using Definiens Tissue studio 2.7 (Definiens, Munich). Airways were annotated manually in each lung based on their morphology to define the regions of interest (ROI) and % of IHC area stained was calculated in each ROI based on a threshold for positive stain for αSMA, Collagen and Fibronectin.

## Statistical analysis and reproducibility

Data is represented as the mean ± standard error of the mean (SEM). All statistical tests were carried out using Prism package (GraphPad software). Student's T-test was performed when comparing two groups for statistical analysis. Analysis of variance (ANOVA) with Dunnett's multiple comparisons post hoc test was used for multiple comparisons for one-way ANOVA and Tukey's post hoc for two-way ANOVA. Distributions of datasets were routinely tested in GraphPad Prism using normality and Lognormality analysis. We observed normally distributed data in our study. Statistically significant values were taken as $*p < 0.05$; $**p < 0.01$; $***p < 0.001$; $****p < 0.0001$ and were assigned in specific figures and experiments as shown. Multiple independent experiments were conducted as detailed in figure legends and showed similar findings. Data from single images or experiments are representative of overall findings from all experiments.

## Reporting summary

Further information on research design is available in the Nature Research Reporting Summary linked to this article.

# Data availability

Complete BioID analysis datasets are provided in Supplementary Data 1.

The BioID mass spectrometry proteomics data have been deposited to the ProteomeXchange Consortium via the PRIDE partner repository with the dataset identifier PXD029237. Source data are provided with this paper.

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

## Acknowledgements

The authors gratefully acknowledge funding from the Medical Research Council UK (MR/S009191/1 and R151002, both to M.P.), the DZHK (to M.G.) and MRC & Asthma UK Centre grant (G1000758) to G.W.). G.S. would like to thank the Biomedical Research Centre Guy's & St Thomas' NHS Foundation Trust and King's College London and a grateful patient for financial support. We would also like to thank Ismael Ranz for assistance with FACS analysis, Dr Claudia Owczarek for assistance with tissue processing, Dr Mark Rigby (Nikon UK) for assistance with image analysis and Janine Fröhlich for support with the BioID experiments.

## Author contributions

M.P., G.S. and E.O.Z. conceived the study. E.O.Z. performed most of the experiments with assistance from D.C.B. and J.R. for ex vivo PCLS imaging, V.L.H. for co-IP and immunofluorescence analysis, L.B.R. for tamoxifen control experiments, flow cytometry and cell recovery, F.V., P.M., M.K. and M.G. all performed and analysed BioID data, IP-M analysed and quantified IHC images, T.J.A.M. and G.W. performed the myography experiments. M.P. and E.O.Z. wrote the paper, with input from all authors. All authors reviewed and approved the final paper.

## Competing interests

The authors declare no competing interests.

## Additional information

Elena Ortiz-Zapater [1,2], Dustin C. Bagley[1], Virginia Llopis Hernandez[1], Luke B. Roberts [3], Thomas J. A. Maguire [4], Felizia Voss[5,6], Philipp Mertins [7], Marieluise Kirchner [7], Isabel Peset-Martin[8], Grzegorz Woszczek [3], Jody Rosenblatt [1], Michael Gotthardt [5,7,9], George Santis[2,10] & Maddy Parsons [1] ✉

[1]Randall Centre for Cell & Molecular Biophysics, King's College London, London, UK. [2]Peter Gorer Department of Immunobiology, School of Immunology and Microbial Science King's College London, London, UK. [3]School of Immunology and Microbial Sciences, King's College London, London, UK. [4]Department of Infectious Diseases, School of Immunology & Microbial Sciences, King's College London, London, UK. [5]Max-Delbrück-Centrum für Molekulare Medizin in the Helmholtz Assoziation (MDC), Berlin, Germany. [6]DZHK Partner site Berlin, Berlin, Germany. [7]Berlin Institute of Health at Charité, Universitaetsmedizin Berlin, Max Delbrück Center for Molecular Medicine (MDC), Berlin, Germany. [8]Medicines Discovery Catapult, Alderley Park, Macclesfield, Cheshire, UK. [9]Charité Universitätsmedizin Berlin, Berlin, Germany. [10]Department of Respiratory Medicine, Guy's & St Thomas NHS Trust, London, UK. ✉e-mail: maddy.parsons@kcl.ac.uk

