## [Peer Review File · Nature Communications]

Epithelial Coxsackievirus Adenovirus Receptor promotes house dust mite-induced lung inflammationREVIEWER COMMENTS

Reviewer #1 (Remarks to the Author):

Ortiz-Zapater and co-workers in their current manuscript show that airway epithelial Coxsackievirus Adenovirus receptor (CAR) expression has a dual role in house dust mite-induced manifestations of asthma, facilitating epithelial pro-inflammatory activity, infiltration of inflammatory cells and goblet cell hyperplasia, while (as described in the abstract) impeding TGF-beta-mediated effects on smooth muscle proliferation, matrix production and airway hyperresponsiveness. They also show data that support a role for CAR in facilitating airway remodeling. While the data certainly may be of interest to a specific audience, I have some concerns and questions that need to be addressed before the manuscript is considered further.

Major comments:

- 1) It would be helpful if the authors could describe whether tamoxifen treatment affected lung morphology or airway inflammation? Also, authors need to describe the Mock mice better. Were these mice also treated with tamoxifen?
- 2) Authors use parametric testing throughout the manuscript, but do not describe whether they tested for normal distribution of the data?
- 3) Were differences in H&E staining in Fig. 1A significant and how was PAS intensity quantified?
- 4) Did authors test for differences in any of the readouts between PBS treated mock and KO mice? This would be important to know in order to determine whether the effects are specific for HDM treatment.
- 5) Were levels of IL1beta and CCL20 significantly higher at baseline in CAR KO vs parental 16HBE cells? In contrast to the beneficial effects of CAR depletion suppressing HDM-induced epithelial pro-inflammatory responses, CAR depletion seems to result in increased pro-inflammatory activity at baseline. Similarly, CAR depletion resulted in increased epithelial permeability at baseline. It would be of interest to know whether increased permeability to dextran paralleled by loss of electrical resistance? Authors mention the reduced barrier integrity, but do not show these data. Furthermore, previous studies have shown that loss of epithelial integrity is accompanied by increased pro-inflammatory activity. Authors should relate their data to these previous findings. Moreover, how can authors explain that CAR KO increases epithelial permeability, with the effect of HDM being no longer present, while reintroduction of CAR reduces epithelial permeability but does not prevent the effect of HDM? Could it be that CAR expression has a protective effect at baseline when in non-phosphorylated form, but upon HDM exposure CAR signaling is modulated (e.g. by phosphorylation) to interfere with its protective effects and/or to promote detrimental/disruptive effects? This explanation is not clearly described in the abstract, the introduction and first part of the results section, but is in line with the finding that CAR depletion leads to TGF-beta-mediated airway remodeling and with the description in the discussion. It would be helpful to align this better.
- 6) It is unclear to me why precision cut lung slices were used to confirm lower infiltration of inflammatory cells upon HDM exposure in KO mice, as immune cells are not present in this model. This needs explanation. Moreover, it is unclear how long after HDM exposure slices were prepared and for how long these were kept in culture.
- 7) In the experiments using conditioned medium from 16HBE cells with/without CAR expression to induce chemotaxis of HL60 cells, was HDM washed away before harvesting the conditioned media? Did parental 16HBE cells also undergo the complete procedure of CRISPR/Cas generation of the cell line? Importantly, it is not clear from the figure whether HL60 migration was significantly lower at baseline or after HDM treatment in parental versus CRISPR/Cas KO cells?
- 8) While I appreciate that CRISPR/Cas KO in primary epithelial cells is a challenge, it would be of great relevance and translational value to confirm the major findings, including effects of HDM on pro-inflammatory responses and barrier function in primary airway epithelial cells and confirm effects of CAR silencing using siRNA.
- 9) Finally, it would be of relevance to show whether TGF-beta treatment results in higher GSK-beta phosphorylation in CAR KO vs parental 16HBE cells.

Minor:

- 1) The manuscript would benefit from some textual editing, containing various syntax errors. For instance, Introduction "...components of the immune system and epithelial cell play", this should be

“...epithelial cells...” or “...epithelial layer...”.

- 2) In the introduction, what is meant exactly by the mucosal immune system? I would suggest to describe the composition of the airway epithelial layer and its functions in a bit more detail in the introduction. Additionally, the composition of epithelial junctions can be described more clearly and how these regulate pro-inflammatory activity and immune responses (e.g. the crucial role of cell adhesion molecule E-cadherin in formation of tight and adherens junctions as well as regulation of various cellular processes).
- 3) The abstract lacks a description of the Methodology.
- 4) I suggest to show the gating strategy for the flow cytometry analyses in the online data supplement.
- 5) Fig. 3C, how can authors be sure that specifically adhesion to collagen is impaired in CAR KO epithelial cells?
- 6) It is not fully clear to me which of the factors shown in Fig. 2B were significantly different between parental and KO cells exposed to HDM and why authors focus on IL-19 here.
- 7) Caveolin-1 expression at epithelial junctions has previously been demonstrated not only to stabilize E-cadherin-mediated junctions, but also to suppress pro-inflammatory/Th2-driving responses in airway epithelial cells, which would be useful to mention (or even confirm).

Reviewer #2 (Remarks to the Author):

In this manuscript, Parsons and colleagues present their studies on the role of the coxsackievirus and adenovirus receptor (CAR) in house dust mite-induced lung inflammation. The experimental approach is built on generating double transgenic mice with Tamoxifen-inducible deletion of the CAR gene specifically in club cells of the mouse airways followed by intranasal inoculation of house dust mite (HDM). The results show that CAR plays a dual role during HDM-induced in airway inflammation by controlling leukocyte infiltration and by regulating tissue remodelling including epithelial architecture and TGF- β -induced smooth muscle cell proliferation.

The work is nicely done and presented in a structured way. The results provide novelty and significant advancement in terms of understanding the role of CAR in lung inflammation. They are also in line with previous reports showing a role for CAR in promoting transepithelial migration of leukocytes. The conclusion matches the results although there are several important issues that need to be answered.

- 1) The usage of tamoxifen for the induction of CAR KO in the airway epithelium. The method section describing how and when tamoxifen was used lacks details. For one thing, it is not clear how long the period was between the ip injection of tamoxifen and HDM inoculation. It is also not clear whether the control animals were given tamoxifen. These issues are critical to answer as tamoxifen by itself is known to have effects on inflammation. Thus, it is important to that the control mice are injected with tamoxifen to ensure that the effects seen in the CAR KO mice are not caused by tamoxifen.
- 2) Differences in leukocyte counts in BAL fluid and tissues. It is unclear why the differences in the infiltration of leukocytes into the airways of CAR KO animals vs control mice does not come out in cellular counts of the BAL fluids. Flow cytometry should be used to study individual immune cell populations in the BAL fluids.
- 3) The findings showing that CAR regulates GSK3 β -TGF β signaling and smooth muscle proliferation. These results are intriguing and relate to other recent findings showing that CAR controls the AKT/GSK3 β signaling pathway and TGF β -induced EMT in breast cancer cells (Nilchian et al, Cancer Res. 2019; 79:1). Please discuss the findings in this manuscript and how they may relate to an overall role of CAR in regulating the activity of these pathways and the relation to epithelial architecture and tissue remodeling.
- 4) Page 3; 2nd paragraph; “Significantly reduced neutrophil infiltration in Ep-CAR KO animals...”; This refers to Supp Fig 2B, which is not showing neutrophils.
- 5) Discussion: end of first paragraph; “Thus, whilst CAR is essential for maintenance of adult lung architecture it also has a negative impact by promoting lung inflammation.” This statement argues that an inflammatory response is negative although in fact, it is essential for the body to deal with pathogens. It would be more appropriate to state that CAR has a dual role in regulating epithelial

architecture and promoting inflammation in the airways.

Reviewer #3 (Remarks to the Author):

This comprehensive and provocative manuscript describes the pivotal role CAR plays in airway structural biology as well as its role in allergic inflammation induced by house dust mite exposure. It carefully dissects various interacting partners and weaves together a complex set of pathways that play roles in epithelial tight junction structure, neutrophil and T-cell recruitment, and smooth muscle and fibroblast proliferation. Positive and negative controls along with unbiased approaches to understand this network strengthen the basis of the models presented. This is important work that describes how a transmembrane protein can have an important role integrating multiple signals and even touches on how pathogens have co-opted essential proteins for cell entry. There are several concerns and suggestions that would strengthen this work.

Page 2 – Supplemental Fig 1. Is 18S the ribosomal RNA subunit? How is CAR ~10x higher relative to such an abundant RNA? Please clarify.

Page 3 – ‘generated rescue cell lines by stably infecting with CAR-GFP’ – the methods are not clear here or in the methods section. How are these cell lines created and are they polyclonal or from a single selected clone? Do you have data in more than one cell line?

IL-6 has previously been shown to be produced by airway cells and acts a chemoattractant. Please speculate why this cytokine either is overlooked or why it is not found in these studies.

In Fig 2B, the scale is row min/max – it is unclear what this means. Is it relative to one of the members of the row or is it based on an absolute number? It is interesting that IL-10 is highly upregulated in this array in CAR-CRISPR + HDM. Please speculate about the role this highly anti-inflammatory cytokine may play in suppressing the inflammatory response.

Some supplementary figure are incorreced referenced.

Page 3 Supp Fig 2B should be 2C. Supp Fig 3C should be 3D.

Page 3 – sentence ‘The reduced MMP9 secretion in the absence of CAR after HDM treatment was significantly reduced using gelatin degradation assays in HDM-treated CAR CRISPR cells compared to parental controls’ is difficult to understand. Please clarify that elevated MMP9 degrades gelatin and CAR is not degrading or inhibiting the degradation of gelatin.

Page 4 – ‘significant increase in epithelial cell height’ – is this the individual cells or stratification or epithelial thickness as referenced after this? It is impossible to see that the individual cells are taller from the figure if this is what is actually meant. There appears to be trends to increased thickness. Please clarify in methods how this was measured and for all references to significance, please clarify the comparison. It is not sufficient to just say something is significant – it must be relative to something.

Page 4 – it is recommended that data in Supp Fig 4 is added to Fig 3. It is important data that should be in the main paper. Note also that green and cyan are not good contrasting colors and other colors should be chosen.

Page 4 – Fig 4F and G are very difficult to understand unless highly trained in the method. Please clarify what the data is showing and how it is interpreted.

Page 4 – Please clarify the BioID – CAR-BirA methods/experiment. Where is the tag added? Between ‘the PDZ-binding domain and the transmembrane domain’ is a very large region that could alter interactions dramatically. Also, it is not clear how CAR was ‘put back in’. What are the possible interactions that can be missed or changed based on where the tag is added? Similarly, where is the GFP tag on CAR. What interactions could be altered based on its location?

Page 6 – Fig 8A uses a blue-white scale – why not a red-blue scale? Is the magnitude different than that of red-blue? Fig 8B – clarify ‘CRISPR’ means CRISPR CAR’.

Page 9 – What dose ‘snapped’ mean?

General:

Minor editing throughout.

The meaning of your replicates are not clear. How do you get multiple airways from a single mouse? Please clarify the experiments that use explants versus tissue sections. Is 5 airways if on a section sufficiently representative? It would be better to specify the size of airway being examined since the

'airway' size can vary widely between, large, medium and small airways.
Were only female mice used in these studies? If so, what is the justification?

We would like to thank the reviewers for their careful and constructive consideration and evaluation of our manuscript. We have responded to each point in turn below; our responses are in blue font for clarity.

Reviewer #1

Ortiz-Zapater and co-workers in their current manuscript show that airway epithelial Coxsackievirus Adenovirus receptor (CAR) expression has a dual role in house dust mite-induced manifestations of asthma, facilitating epithelial pro-inflammatory activity, infiltration of inflammatory cells and goblet cell hyperplasia, while (as described in the abstract) impeding TGF-beta-mediated effects on smooth muscle proliferation, matrix production and airway hyperresponsiveness. They also show data that support a role for CAR in facilitating airway remodeling. While the data certainly may be of interest to a specific audience, I have some concerns and questions that need to be addressed before the manuscript is considered further.

Major comments:

1) It would be helpful if the authors could describe whether tamoxifen treatment affected lung morphology or airway inflammation? Also, authors need to describe the Mock mice better. Were these mice also treated with tamoxifen?

We apologise if this was unclear. As we describe in the methods section of the original submission, 'Mock' mice were those carrying the CAR^{fl/fl} allele but with IP injection of corn oil, not tamoxifen. All experiments were initiated 2 weeks after the final injection of either agent. We chose this as a control to ensure mice carried identical genetic backgrounds across all experiments for better comparison. When optimising the knockout conditions, we profiled lungs by qPCR (for CAR expression) and H&E at day 7 post-Tamoxifen administration and saw no changes in immune cell infiltrates or lung tissue organisation in tamoxifen treated animals compared to corn oil treated controls ('Mock'). We also note that as shown in our manuscript, EpCAR^{-/-} mice treated with Tamoxifen show no change in immune response to HDM in the BAL compared to control animals; the key changes we observe to specific immune cell populations occur at the level of immune cell lung tissue infiltration further suggesting a lung-specific CAR-dependent response. However, to further confirm that Tamoxifen does not drive an inflammatory response in the lung that may contribute to interpretation of our findings, we have now performed additional experiments administering Tamoxifen to C57B/6J mice (the background on which our EpCAR^{-/-} model was generated) compared to corn oil only (the vehicle used for Tamoxifen), administering IP for one week (identical to our EpCAR^{-/-} protocol) followed by sacrificing animals 2 weeks later (the point at which we would usually administer HDM or PBS). Inspection of the lungs stained with H&E and formal quantification of immune profiles in the lungs using FACS analysis both demonstrated no change in organisation or inflammatory profiles. Importantly with respect to the reviewers' comments, the immune populations that we see altered in the lungs of EpCAR^{-/-} mice (CD45⁺, neutrophils, $\gamma\delta$ Tcells) are unaffected by Tamoxifen treatment, providing additional confidence that the data we present reflects a specific CAR-dependent phenotype. We have included this data in Supp Figure 1D and E of our revised manuscript.

2) Authors use parametric testing throughout the manuscript, but do not describe whether they tested for normal distribution of the data?

We apologise for this omission. We routinely test distribution of datasets using GraphPad prism. We did not note any trends away from normally distributed data in our study. We have included this in the statistical analysis section of the methods in our revised manuscript.

3) Were differences in H&E staining in Fig. 1A significant and how was PAS intensity quantified?

We did not quantify the H&E sections as the subsequent data we show provides greater depth of information with respect to quantified specific subpopulations of cells and key markers which we feel the H&E sections alone cannot provide. The PAS data was quantified as described in the methods section of our manuscript ("Airways were annotated manually in each lung based on their morphology to define the regions of interest (ROI) and % of IHC area stained was calculated in each ROI based on a threshold for positive stain").

4) Did authors test for differences in any of the readouts between PBS treated mock and KO mice? This would be important to know in order to determine whether the effects are specific for HDM treatment. Yes, we performed statistical analysis on all data from Mock vs EpCAR KO throughout and statistical differences are shown where they occur throughout allowing us to form conclusions around changes relating to depletion of CAR or those induced by HDM. Specifically, the data in Figures 3, 6 and 8 focus on the basal changes (non-HDM induced) in Mock vs EpCAR KO animals – with additional comparisons in Figure 6C and E demonstrating baseline changes in Mock vs EpCAR KO animals in airway size and smooth muscle actin assembly which are unchanged in EpCAR KO mice upon HDM treatment. We additionally performed the same level of analysis across conditions for the *in vitro* human epithelial cell culture work throughout our study.

5) Were levels of IL1beta and CCL20 significantly higher at baseline in CAR KO vs parental 16HBE cells? In contrast to the beneficial effects of CAR depletion suppressing HDM-induced epithelial pro-inflammatory responses, CAR depletion seems to result in increased pro-inflammatory activity at baseline. Similarly, CAR depletion resulted in increased epithelial permeability at baseline. It would be of interest to know whether increased permeability to dextran paralleled by loss of electrical resistance? Authors mention the reduced barrier integrity, but do not show these data. Furthermore, previous studies have shown that loss of epithelial integrity is accompanied by increased pro-inflammatory activity. Authors should relate their data to these previous findings. Moreover, how can authors explain that CAR KO increases epithelial permeability, with the effect of HDM being no longer present, while reintroduction of CAR reduces epithelial permeability but does not prevent the effect of HDM? Could it be that CAR expression has a protective effect at baseline when in non-phosphorylated form, but upon HDM exposure CAR signalling is modulated (e.g. by phosphorylation) to interfere with its protective effects and/or to promote detrimental/disruptive effects? This explanation is not clearly described in the abstract, the introduction and first part of the results section, but is in line with the finding that CAR depletion leads to TGF-beta-mediated airway remodelling and with description in the discussion. It would be helpful to align this better.

We thank the reviewer for these points and apologise if they felt we did not articulate the proposed CAR-dependent control of airway function in the manuscript. We did not assess IL1-b or CCL20 levels in the *in vitro* model as we did not see changes to these cytokines at baseline level (without HDM) in the mouse model (Figure 2A of the original manuscript). We were unsure as to the data the reviewer referred to with respect to increased pro-inflammatory cytokine release upon CAR depletion. The data in Figures 2A-D demonstrate no changes in control vs CAR KO mice or cells basally – increased cytokine release is seen in control mice/cells upon HDM treatment, and this induction is inhibited when CAR is depleted. We did not assess TER in the *in vitro* model; however, CAR has previously been shown to positively regulate TER in several previous publications (e.g.: PMID: 11734628; PMID: 21918008; PMID: 16413486) and that TER directly correlates with dextran permeability (e.g. as shown in PMID: 16413486) and thus either assay can report on barrier integrity. In response to the reviewers point around the permeability assays we show: our data is consistent with the role of CAR in regulating HDM-induced permeability, in that rescue of CAR CRISPR cells with WTCAR leads to the same increased permeability as seen in parental epithelial cells (Fig 3B). This is exactly what would be expected and confirms that CAR-dependent functions in the CAR CRISPR cells can be restored by re-expressing WTCAR (as we also see in Fig 1I). However, we agree entirely that some other studies have shown correlations between loss of barrier integrity and increased inflammatory signals from the epithelium, the latter of which we do not see when we deplete CAR. As the reviewer points out, we provide some discussion and potential explanations for this in our manuscript (paragraphs 2 and 3, page 7 of the original submission). We postulate that phosphorylation of CAR induced by pro-inflammatory triggers (e.g.: HDM) destabilises this receptor at the membrane and induces the pro-inflammatory cues we show here, as well as promoting CAR-immune cell interactions to induce inflammation. As the reviewer suggests, we have now included further mention of this concept at the end of the introduction to better reflect this earlier on in the manuscript.

6) It is unclear to me why precision cut lung slices were used to confirm lower infiltration of inflammatory cells upon HDM exposure in KO mice, as immune cells are not present in this model. This needs explanation. Moreover, it is unclear how long after HDM exposure slices were prepared and for how long these were kept in culture.

PCLS have previously been used by a number of other groups to study immune cell behaviour in different organisms, including macrophages, neutrophils, DCs and T cells (PMID: 31924278; PMID: 31175176; PMID: 28448013). PCLS have also been used for studies of inflammatory responses to host–pathogen interactions, including viral and bacterial infection [PMID: 19913271; PMID: 23185463; PMID: 15047831; PMID: 24712747; PMID: 25916988]. Furthermore, PCLS are applicable to live, dynamic imaging of immune interactions (PMID: 28448013; PMID: 24478099). Thus, PCLS are recognised as a suitable model for studying immune cell behaviour and tissue interactions.

We apologise if the details of PCLS preparation were unclear. PCLS were generated 24 hours following the final HDM inhalation dose. Lung slices were prepared and if used for fixed immunofluorescence, were fixed in 4% PFA on the same day as preparation. For live experiments, PCLS were placed overnight in growth medium as described in the Methods section and imaged the following day. PCLS were viable and Methacholine-responsive for up to 12 days post-isolation.

7) In the experiments using conditioned medium from 16HBE cells with/without CAR expression to induce chemotaxis of HL60 cells, was HDM washed away before harvesting the conditioned media? Did parental 16HBE cells also undergo the complete procedure of CRISPR/Cas generation of the cell line? Importantly, it is not clear from the figure whether HL60 migration was significantly lower at baseline or after HDM treatment in parental versus CRISPR/Cas KO cells?

For the experiments shown in Figure 1I of the original submission, 16HBE cells were treated with HDM for 24h, and conditioned media (containing HDM where indicated) was taken directly and used in the HL-60 transwell assays. HL60 migration was significantly lower in both basal (untreated) and HDM treated conditions compared to respective parental controls. We apologise for the missing statistics referring to baseline differences; these have been added into the revised manuscript Figure 1I.

8) While I appreciate that CRISPR/Cas KO in primary epithelial cells is a challenge, it would be of great relevance and translational value to confirm the major findings, including effects of HDM on pro-inflammatory responses and barrier function in primary airway epithelial cells and confirm effects of CAR silencing using siRNA.

We agree this would be an interesting complementary set of experiments to compare to our in vivo and in vitro studies. However, as the reviewer points out, working with primary airway epithelial cells is extremely challenging. Indeed, we have tried to culture primary cells from the CAR-Epfl/fl mice in vitro and deplete CAR using siRNA from primary human cells without success; the mouse primary cells only grew for one passage on feeders and then died; the primary cell transfection efficiency was poor and efforts to improve this (using electroporation, for example) resulted in high levels of toxicity. However, we feel that the current data showing highly correlated findings using murine in vivo and ex vivo cultures as well as human cell lines are sufficient to provide confidence that our findings are robust and biologically relevant.

9) Finally, it would be of relevance to show whether TGF-beta treatment results in higher GSK-beta phosphorylation in CAR KO vs parental 16HBE cells.

We were rather unsure as to the rationale for this request. Our data demonstrates that depletion of CAR results in increased TGF β production and pSMAD2/3 (Figure 8E-I, and this is coupled with enhanced pGSK3 β (Figures 8B-D). We further show that enhancing pGSK3 β leads to increased levels of TGF β production by parental cells (Figure 8I), strongly suggesting that pGSK3 β leads to increased TGF β (and downstream ECM synthesis, Figure 8K) not vice versa. Hence, we do not feel that treating CAR KO cells with more TGF β (given that this is already enhanced in these cells) will be informative as our data indicates that the pathway operates in reverse (GSK3 β >TGF β , as opposed to TGF β >GSK3 β).

Minor:

1) The manuscript would benefit from some textual editing, containing various syntax errors. For instance, Introduction "...components of the immune system and epithelial cell play", this should be "...epithelial cells..." or "...epithelial layer...".

We apologise for these errors and have corrected this in the revised manuscript version.

2) In the introduction, what is meant exactly by the mucosal immune system? I would suggest describing the composition of the airway epithelial layer and its functions in a bit more detail in the introduction. Additionally, the composition of epithelial junctions can be described more clearly and how these regulate pro-inflammatory activity and immune responses (e.g. the crucial role of cell adhesion molecule E-cadherin in formation of tight and adherens junctions as well as regulation of various cellular processes).

We thank the reviewer for this suggestion and have added additional detail into the introduction as suggested to provide further background and context to the study in these areas.

3) The abstract lacks a description of the Methodology.

Nature Communications set a maximum word limit of 150 words for the abstract, so unfortunately, we are unable to include further details in the abstract as requested.

4) I suggest showing the gating strategy for the flow cytometry analyses in the online data supplement.

The FACS gating strategy is described in the 'Reporting Summary' document that accompanies our manuscript, along with additional details pertaining to the methods and reagents used in our study.

5) Fig. 3C, how can authors be sure that specifically adhesion to collagen is impaired in CAR KO epithelial cells?

We apologise if this was unclear: cells were plated on coverslips coated with purified type I collagen in Figure 3C (and Matrigel in Figure 3D) enabling us to specifically conclude defects in adhesion to this extracellular matrix protein.

6) It is not fully clear to me which of the factors shown in Fig. 2B were significantly different between parental and KO cells exposed to HDM and why authors focus on IL-19 here.

We apologise for this; we found it rather challenging to know how to include statistical analysis on data presented in this format. However, we have now added asterisks after each cytokine/molecule to indicate cases where we see significantly different responses in CAR KO cells to HDM and have indicated this in the figure legends in the revised manuscript. We hope this clarifies these findings for the reviewer. Due to the large numbers of targets identified in this experiment, it was impractical to follow-up on all targets in functional experiments, and we chose to focus on targets showing differential response in CAR KO mice/cells. As we stated in our manuscript results text (p3, paragraph 3), we chose to focus on IL-19 as recent reports have shown positive correlations between levels of IL-19 and allergic airway inflammation as well as boosting of typical Th2 cytokines such as IL-4 and IL-13. We therefore felt this cytokine represented a strong candidate for co-ordination of the CAR-induced immune response; indeed, our experiments in Fig 2E indicate this is the case. We feel this data provides support to the notion that the altered cytokine profiles seen in CAR KO cells are functionally important in terms of immune response. We have added additional text to the results and discussion sections of the revised manuscript to highlight the other changes seen in vitro and in vivo.

7) Caveolin-1 expression at epithelial junctions has previously been demonstrated not only to stabilize E-cadherin-mediated junctions, but also to suppress pro-inflammatory/Th2-driving responses in airway epithelial cells, which would be useful to mention (or even confirm).

We thank the reviewer for this point. We did include mention of this role for caveolin in our discussion but have expanded on this in the revised version. Whilst we agree this would be interesting to consider further in future, we believe that extensive experiments detailing the molecular basis for the interactions between CAR and caveolin and implications for force sensing and immune response lie beyond the scope

of the current study.

Reviewer #2

In this manuscript, Parsons and colleagues present their studies on the role of the coxsackievirus and adenovirus receptor (CAR) in house dust mite-induced lung inflammation. The experimental approach is built on generating double transgenic mice with Tamoxifen-inducible deletion of the CAR gene specifically in club cells of the mouse airways followed by intranasal inoculation of house dust mite (HDM). The results show that CAR plays a dual role during HDM-induced airway inflammation by controlling leukocyte infiltration and by regulating tissue remodelling including epithelial architecture and TGF- β -induced smooth muscle cell proliferation. The work is nicely done and presented in a structured way. The results provide novelty and significant advancement in terms of understanding the role of CAR in lung inflammation. They are also in line with previous reports showing a role for CAR in promoting transepithelial migration of leukocytes. The conclusion matches the results although there are several important issues that need to be answered:

1) The usage of tamoxifen for the induction of CAR KO in the airway epithelium. The method section describing how and when tamoxifen was used lacks details. For one thing, it is not clear how long the period was between the IP injection of tamoxifen and HDM inoculation. It is also not clear whether the control animals were given tamoxifen. These issues are critical to answer as tamoxifen by itself is known to have effects on inflammation. Thus, it is important that the control mice are injected with tamoxifen to ensure that the effects seen in the CAR KO mice are not caused by tamoxifen.

We apologise if the reviewer felt this was unclear. As stated in the Methods of our original submission "Intraperitoneal administration of Tamoxifen (Sigma) was used to induce the lung epithelial expression of Cre and excision of CAR. Tamoxifen was administered using 75 mg tamoxifen/kg body mouse weight for 6 consecutive days". Where relevant, HDM was first dosed to animals 3 weeks following the initial Tamoxifen injection. We have added the latter information into the methods of our revised manuscript for clarity. For details of the point around Tamoxifen treatment: this was also raised by Reviewer 1, point 1, and we provide our response again here for convenience:

As we describe in the methods section of the original submission, 'Mock' mice were those carrying the CAR^{fl/fl} allele but with IP injection of corn oil, not tamoxifen. All experiments were initiated 2 weeks after the final injection of either agent. We chose this as a control to ensure mice carried identical genetic backgrounds across all experiments for better comparison. When optimising the knockout conditions, we profiled lungs by qPCR (for CAR expression) and H&E at day 7 post-Tamoxifen administration and saw no changes in immune cell infiltrates or lung tissue organisation in tamoxifen treated animals compared to corn oil treated controls ('Mock'). We also note that as shown in our manuscript, EpCAR^{-/-} mice treated with Tamoxifen show no change in immune response to HDM in the BAL compared to control animals; the key changes we observe to specific immune cell populations occur at the level of immune cell lung tissue infiltration further suggesting a lung-specific CAR-dependent response. However, to further confirm that Tamoxifen does not drive an inflammatory response in the lung that may contribute to interpretation of our findings, we have now performed additional experiments administering Tamoxifen to C57B/6J mice (the background on which our EpCAR^{-/-} model was generated) compared to corn oil only (the vehicle used for Tamoxifen), administering IP for one week (identical to our EpCAR^{-/-} protocol) followed by sacrificing animals 2 weeks later (the point at which we would usually administer HDM or PBS). Inspection of the lungs stained with H&E and formal quantification of immune profiles in the lungs using FACS analysis both demonstrated no change in organisation or inflammatory profiles. Importantly with respect to the reviewers' comments, the immune populations that we see altered in the lungs of EpCAR^{-/-} mice (CD45⁺, neutrophils, $\gamma\delta$ Tcells) are unaffected by Tamoxifen treatment, providing additional confidence that the data we present reflects a specific CAR-dependent phenotype. We have included this data in Supp Figure 1D and E of our revised manuscript.

2) Differences in leukocyte counts in BAL fluid and tissues. It is unclear why the differences in the

infiltration of leukocytes into the airways of CAR KO animals vs control mice does not come out in cellular counts of the BAL fluids. Flow cytometry should be used to study individual immune cell populations in the BAL fluids.

We thank the reviewer for raising this. We agree it is very interesting that EpCAR^{-/-} mice are able to mount a broadly equivalent response to HDM as seen in control mice, but specific subpopulations of immune cells are not increased in the lung. Our data would strongly suggest that this is due to the significantly reduced levels of key HDM-induced pro-inflammatory factors produced by the lung epithelium in EpCAR^{-/-} animals following HDM challenge. This would be expected to reduce lung immune cell infiltration, which is indeed what we observe. Moreover, as CAR is a direct receptor for molecules expressed on the surface of specific immune cell populations, loss of CAR would also be expected to contribute to reduced tissue integration of immune cells and interactions with the epithelium. Finally, the increased basal collagen deposition seen in EpCAR^{-/-} mice may further prevent efficient immune cell tissue infiltration and migration. The combination of these factors very likely explains this lack of tissue-level immune cell infiltration. We have added additional text to reflect this into the discussion section of our revised manuscript.

We have included the full immune cell profiles from BAL from all mice and conditions in Supplementary figure 2 (B-D) of our revised manuscript as requested. This data demonstrates that all populations of immune cells we profiled in the BAL from Mock mice (with and without HDM) match those seen in EpCAR^{-/-} mice. This would further suggest as discussed above that CAR plays a role in control of tissue-level inflammation.

3) The findings showing that CAR regulates GSK3b-TGFb signaling and smooth muscle proliferation. These results are intriguing and relate to other recent findings showing that CAR controls the AKT/GSK3b signaling pathway and TGFb-induced EMT in breast cancer cells (Nilchian et al, Cancer Res. 2019; 79:1). Please discuss the findings in this manuscript and how they may relate to an overall role of CAR in regulating the activity of these pathways and the relation to epithelial architecture and tissue remodeling.

We are very grateful to the reviewer for pointing out the potential parallels in terms of signalling pathways identified in this study and ours. We have included reference to the Nilchian et al paper and potential implications of or combined findings in the revised manuscript discussion text.

4) Page 3; 2nd paragraph; “Significantly reduced neutrophil infiltration in Ep-CAR KO animals...”; This refers to Supp Fig 2B, which is not showing neutrophils.

We apologise for this error which has been corrected in the revised manuscript.

5) Discussion: end of first paragraph; “Thus, whilst CAR is essential for maintenance of adult lung architecture it also has a negative impact by promoting lung inflammation.” This statement argues that an inflammatory response is negative although in fact, it is essential for the body to deal with pathogens. It would be more appropriate to state that CAR has a dual role in regulating epithelial architecture and promoting inflammation in the airways.

We thank the reviewer for this suggestion and have amended the text in our revised manuscript accordingly.

Reviewer #3

This comprehensive and provocative manuscript describes the pivotal role CAR plays in airway structural biology as well as its role in allergic inflammation induced by house dust mite exposure. It carefully dissects various interacting partners and weaves together a complex set of pathways that play roles in epithelial tight junction structure, neutrophil and T-cell recruitment, and smooth muscle and fibroblast proliferation. Positive and negative controls along with unbiased approaches to understand this network strengthen the basis of the models presented. This is important work that describes how a transmembrane protein can have an important role integrating multiple signals and even touches on how pathogens have co-opted essential proteins for cell entry. There are several concerns and

suggestions that would strengthen this work:

1) Page 2 – Supplemental Fig 1. Is 18S the ribosomal RNA subunit? How is CAR ~10x higher relative to such an abundant RNA? Please clarify.

We apologise if this was unclear; we used levels of 18S to normalise CAR expression to correct for any potential differences in total cDNA input and the data presented is normalised relative CAR expression across the specified genotypes/treatments.

2) Page 3 – ‘generated rescue cell lines by stably infecting with CAR-GFP’ – the methods are not clear here or in the methods section. How are these cell lines created and are they polyclonal or from a single selected clone? Do you have data in more than one cell line?

We apologise for the lack of detail here. CAR CRISPR cells were infected with a lentivirus encoding WTCAR-GFP (that we have previously characterised and published, as we cite in our methods section). Using this approach, we see ~95% successful infection of the population so additional sorting is not required. Because these are not clonal rescue cells, or even a sorted subset of the original population, we did not feel it was necessary to generate multiple infected populations as these cells were essentially used to confirm successful restoration of CAR CRISPR phenotypes mitigating for any potential off-target effects of the CRISPR Cas-9 depletion. We have added additional details on generation of the WTCAR-GFP CRISPR rescue cells in the revised manuscript methods section.

3) IL-6 has previously been shown to be produced by airway cells and acts a chemoattractant. Please speculate why this cytokine either is overlooked or why it is not found in these studies.

We agree that IL-6 is an important component of the epithelial response and has been implicated in allergic airway inflammation. Interestingly, we did not see a reduction in HDM-induced IL-6 production in CAR depleted cells (Figure 2B) indicating that CAR does not play a role in regulating this cytokine. As also articulated in the response to point 4 below, due to the large numbers of targets identified in this experiment, it was impractical to follow-up on all targets in functional experiments, and we chose to focus on targets showing differential response in CAR KO mice/cells. The lack of differential response in IL-6 may indicate that CAR plays a more direct role in controlling a Th2-like response, although noting that the precise mechanisms by which this is regulated remain unknown and will be the focus of future studies that lie beyond the scope of the current manuscript.

4) In Fig 2B, the scale is row min/max – it is unclear what this means. Is it relative to one of the members of the row or is it based on an absolute number? It is interesting that IL-10 is highly upregulated in this array in CAR-CRISPR + HDM. Please speculate about the role this highly anti-inflammatory cytokine may play in suppressing the inflammatory response.

We apologise for the lack of clarity on the data presentation in figure 2B. Each row (representing a different cytokine/molecule) is scaled to show the smallest (min, dark blue) and maximum (max, red) densitometry readings for that molecule, meaning each row is scaled independently of the others in terms of absolute values. The data is presented in this manner due to the very large differential in the levels of each molecule represented on the array and provides means to evaluate and display the changes in expression levels of each target across the different conditions (which was the goal of this experiment). We have clarified this in the methods section of our revised manuscript. We agree that the increased levels of IL-10 in CAR KO cells + HDM is interesting and may potentially contribute to the anti-inflammatory effects seen in CAR KO conditions. Due to the large numbers of targets identified in this experiment, it was impractical to follow-up on all targets in functional experiments, but we are interested in pursuing this in future studies. As suggested by the reviewer, we have added some additional text to the revised manuscript to highlight the other changes seen in this array and speculate on their potential roles.

5) Some supplementary figures are incorreced referenced. Page 3 Supp Fig 2B should be 2C. Supp Fig 3C should be 3D.

We apologise for these errors and have corrected these in the revised manuscript.

6) Page 3 – sentence ‘The reduced MMP9 secretion in the absence of CAR after HDM treatment was significantly reduced using gelatin degradation assays in HDM-treated CAR CRISPR cells compared to parental controls’ is difficult to understand. Please clarify that elevated MMP9 degrades gelatin and CAR is not degrading or inhibiting the degradation of gelatin.

We apologise for the confusing wording of these findings and have amended this text as suggested in the revised manuscript.

7) Page 4 – ‘significant increase in epithelial cell height’ – is this the individual cells or stratification or epithelial thickness as referenced after this? It is impossible to see that the individual cells are taller from the figure if this is what is actually meant. There appears to be trends to increased thickness. Please clarify in methods how this was measured and for all references to significance, please clarify the comparison. It is not sufficient to just say something is significant – it must be relative to something.

We apologise if this was unclear. The data shown in Supp Fig 4B is the measurement of the height of the epithelium (apical to basal surface) from images as shown in Supp Fig 4A; the black lines on these images show how measurements were taken (we apologise that mention of these marker arrows was omitted from the legend and has been added to the revised manuscript version). We have also changed the text and figure axis labels to be more consistent in our language when describing this data. We have additionally included more specific details of comparisons of significance as shown in the graphs in Supp Fig4B.

8) Page 4 – it is recommended that data in Supp Fig 4 is added to Fig 3. It is important data that should be in the main paper. Note also that green and cyan are not good contrasting colors and other colors should be chosen.

We agree that the data in Supp Figure 4 is important to support the overall narrative of our study. However, as we describe in the manuscript text, we and others have previously reported that CAR contributes to epithelial cell integrity and as such, we felt this data was confirmatory of previous findings in our model systems, rather than entirely novel findings. For this reason, we believe this data is more appropriate to be left as supplementary material that supports the overall findings as opposed to being embedded in the main figures. We apologise for the lack of clarity in the images and have changed these to green and magenta in-line with our other figures.

9) Page 4 – Fig 4F and G are very difficult to understand unless highly trained in the method. Please clarify what the data is showing and how it is interpreted.

We apologise if the details of these experiments in Figs 4E-G were unclear. Lines of equal length and width were drawn perpendicular to junctions identified by localisation of the target molecule and the intensity values of each channel were normalised to control (PBS) values. This provides means to determine the ‘sharpness’ or presence of the localisation of each molecule to the cell-cell adhesion site, where reduced peaks indicate more diffuse localisation of the target molecule. We have added further text into the results and methods sections of our revised manuscript to better explain this approach and interpretation.

10) Page 4 – Please clarify the BioID – CAR-BirA methods/experiment. Where is the tag added? Between ‘the PDZ-binding domain and the transmembrane domain’ is a very large region that could alter interactions dramatically. Also, it is not clear how CAR was ‘put back in’. What are the possible interactions that can be missed or changed based on where the tag is added? Similarly, where is the GFP tag on CAR. What interactions could be altered based on its location?

We apologise for the lack of detail on the cloning in our original manuscript. The TurboID tag was inserted between Lys285 and Ser286 after a stretch of prolines, to reduce the chance of interference with any essential structural feature. The plasmid was transfected into 16HBE CAR CRISPR cells using Lipofectamine 3000 and we verified localization to the plasma membrane and restoration of cell-cell adhesion in CAR CRISPR cells, indicating proper maturation and trafficking of the CAR-TurboID fusion protein. We have added this information into the Methods section of our revised manuscript. We also note our BioID analysis identified all previously published CAR-interacting partners (denoted in

Supplementary Table 1) providing further confidence that our construct is not altering protein interactions. The CAR-GFP constructs we used here are in a lentiviral backbone that we have previously published and characterised, as cited in the methods section of the original submission. We and others have extensively characterised these CAR-GFP constructs previously as being functional (e.g.: PMID: 24096322; PMID: 29382784; PMID: 27193388; PMID: 24273169; PMID: 19527712), and as we further demonstrate in the current study, expression of WTCAR-GFP is capable of rescuing function in CAR CRISPR cells further supporting the validity of these plasmids.

11) Page 6 – Fig 8A uses a blue-white scale – why not a red-blue scale? Is the magnitude different than that of red-blue? Fig 8B – clarify ‘CRISPR’ means CRISPR CAR’.

We presume the reviewer is referring to the differences in presentation of data between this figure (8A) and that shown in Figure 2B (cytokine array). These are entirely different assay types (the first is ELISA based, the second is a phospho-signalling array based on 1D blotting), so we did not feel it was necessarily required to present them using identical formats. Indeed, presenting them differently perhaps more clearly indicates the fact that these are entirely different types of experiments. We do not feel that the presentation of data as blue/white in Figure 8A is inappropriate, so would prefer to retain this format if the reviewer agrees. We have relabelled the panel in Fig 8B to CAR CRISPR as suggested.

12) Page 9 – What dose ‘snapped’ mean?

We apologise, this was a typo error and should have read ‘tubes were mixed in between sonication’. We have corrected this in the revised manuscript.

13) The meaning of your replicates are not clear. How do you get multiple airways from a single mouse? Please clarify the experiments that use explants versus tissue sections. Is 5 airways if on a section sufficiently representative? It would be better to specify the size of airway being examined since the ‘airway’ size can vary widely between, large, medium and small airways.

We apologise if this was unclear. Where we quantified individual airway sizes or features of therein, the numbers we provided refer to individual airways within PCLS tissue slices (as the reviewer correctly notes, there are multiple airways in each slice). For every experiment we analysed multiple airways per PCLS section from more than one PCLS per animal and across multiple different experiments. We believe this is entirely robust in terms of providing broad representation across multiple samples and mice for statistical analysis. We have further clarified these details in the figure legends of the revised manuscript where appropriate. We did not select airways based on size; all airways that were complete and visible within fields of view for each experiment were included in the analysis.

14) Were only female mice used in these studies? If so, what is the justification?

We performed initial characterisation experiments in both male and female C57BL/6 control mice and found higher variability in the males in terms of immune cell populations in the BAL and tissue following HDM challenge compared to females. Whilst we are unsure as to the reason for this (and have not seen it reported specifically in the literature, although we note many others using this model also use female mice), we felt unknown confounding factors from the use of mixed sex populations may make it difficult to interpret data, so we chose to perform all subsequent experiments in female mice. We have specified this in the methods section of the revised manuscript.

REVIEWER COMMENTS

Reviewer #1 (Remarks to the Author):

The manuscript has been significantly improved, however, a couple of issues remain, which should be addressed.

1) In response to my question regarding the normal distribution of data, authors mention "We did not observe any trends away from normally distributed data in our study". What does this mean, where data normally distributed or not?

2) In the legend of Fig. 1A, please indicate of how many mice and images per mouse data are representative.

3) In their previous figure 2A, the levels of IL-1beta and CCL20 are higher in Ep-CAR KO + PBS mice than in the Mock + PBS mice. This may not have been significant, but it would still be worthwhile to assess levels of these cytokines in 16HBE cells upon knockdown of CAR. Similar for TER measurements, although I appreciate that this may be difficult if authors no longer have the cells in culture.

4) Can the authors exclude that HDM may have affected the migration of HL-60 cells? Some discussion on this would be welcome.

5) The authors should describe their efforts to downregulate siRNA in primary cells in the Methods and show data on their failed attempt (e.g. in the online data supplement).

6) The authors should add the extra information on the PCLS procedure provided in the letter also in the manuscript.

Reviewer #2 (Remarks to the Author): Reviewer #2 was not able to produce a report, so their comments were mediated by reviewer #1.

The authors made substantial effort to address the comments and most issues have now been addressed satisfactorily.

With respect to the first comment on the treatment with Tamoxifen, the authors have shown that Tamoxifen does not induce alterations in the immune response to HDM in the BAL of EpCAR^{-/-} mice, indicating a lung specific effect. In addition, they have now added data on the lack of effect of Tamoxifen at baseline in control animals. However, authors can still not exclude that this effect is due to a combination of Tamoxifen and HDM treatment, as control animals did not receive Tamoxifen. This should be mentioned in the Discussion.

With respect to comment 3, the authors have now included the reference to the Nilchian paper (ref 63), but no alterations in the text have been indicated. Authors should make sure to discuss how regulation of GSK3b-TGF-beta signaling relates to the role of CAR in regulation of epithelial remodeling.

Reviewer #3 (Remarks to the Author):

No further concerns

We thank the reviewers for their time in re-reviewing our revised manuscript and for their additional comments and suggestions which we address below:

Reviewer #1:

The manuscript has been significantly improved, however, a couple of issues remain, which should be addressed.

1) In response to my question regarding the normal distribution of data, authors mention "We did not observe any trends away from normally distributed data in our study". What does this mean, where data normally distributed or not?

We apologise if this was unclear; the data was normally distributed and we have clarified this in the relevant text in the methods.

2) In the legend of Fig. 1A, please indicate of how many mice and images per mouse data are representative.

This information was already provided in the figure legend referring to the quantification of this representative data (Figure 1B).

3) In their previous figure 2A, the levels of IL-1beta and CCL20 are higher in Ep-CAR KO + PBS mice than in the Mock + PBS mice. This may not have been significant, but it would still be worthwhile to assess levels of these cytokines in 16HBE cells upon knockdown of CAR. Similar for TER measurements, although I appreciate that this may be difficult if authors no longer have the cells in culture.

As we mentioned in our previous response to this point, we did not assess IL1- β or CCL20 levels in the *in vitro* model as we did not see significant changes to these cytokines at baseline level (without HDM) in the mouse model in any of our experiments. Indeed, the data in Figures 1 and 2 also demonstrate no changes in control vs CAR KO mice or cells basally in terms of immune cell infiltration or cytokine release. Thus, we remain entirely unsure as to the rationale for performing these experiments or how these experiments might be used in, or contribute to, the rest of our study. We did not assess TER in the *in vitro* model; however, as we also commented in our previous response document, CAR has previously been shown to positively regulate TER in several previous publications (e.g.: PMID: 11734628; PMID: 21918008; PMID: 16413486) and TER directly correlates with dextran permeability (e.g. as shown in PMID: 16413486) and thus either assay can report on barrier integrity. Thus, we also do not feel TER experiments would provide any additional insight to our study.

4) Can the authors exclude that HDM may have affected the migration of HL-60 cells? Some discussion on this would be welcome.

We thank the reviewer for raising this and have added additional text to the results section (p3) of the revised manuscript to highlight this possibility.

5) The authors should describe their efforts to downregulate siRNA in primary cells in the Methods and show data on their failed attempt (e.g. in the online data supplement).

We were unclear as to the rationale for this request; we do not show any primary airway epithelial cell data in our manuscript and as such, we do not feel it is appropriate to include this information in the methods (given that it does not relate to any presented data). As we explained in our previous response document, the attempts to reduce CAR in primary airway cells resulted in high toxicity, so we are also not convinced that showing any data relating to these experiments is appropriate given they were technically highly flawed and are not valuable to the overall manuscript, or in guiding readers in this approach (given that were unsuccessful).

6) The authors should add the extra information on the PCLS procedure provided in the letter also in the manuscript.

We have added the missing PCLS information to the relevant methods sections of the revised manuscript as requested.

Reviewer #2:

The authors made substantial effort to address the comments and most issues have now been addressed satisfactorily.

1) With respect to the first comment on the treatment with Tamoxifen, the authors have shown that Tamoxifen does not induce alterations in the immune response to HDM in the BAL of EpCAR^{-/-} mice,

indicating a lung specific effect. In addition, they have now added data on the lack of effect of Tamoxifen at baseline in control animals. However, authors can still not exclude that this effect is due to a combination of Tamoxifen and HDM treatment, as control animals did not receive Tamoxifen. This should be mentioned in the Discussion.

We thank the reviewer for this suggestion. We have added this comment into the first paragraph of the results section where we describe the inflammation phenotype data from the vivo model as we felt this was the most appropriate place to draw attention to this point.

2) With respect to comment 3, the authors have now included the reference to the Nilchian paper (ref 63), but no alterations in the text have been indicated. Authors should make sure to discuss how regulation of GSK3b-TGF-beta signaling relates to the role of CAR in regulation of epithelial remodeling.

We apologise, the text referring to this paper was in the discussion of the revised manuscript but for some reason was not marked up in the PDF version. This is now visible in the resubmitted manuscript version (page 8, first paragraph). We have added some additional text here as requested to suggest the enhanced TGF β seen in CAR KO cells/mice may play a dual role in control of ECM synthesis and also an EMT-like state within the epithelium (as seen in the Nilchian paper in cancer cells).

Reviewer #3:

No further concern